# Generative Learning of Continuous Data by Tensor Networks

**Alex Meiburg[1,2,3], Jing Chen[1]\*, Jacob Miller[1]†, Raphaëlle Tihon[4], Guillaume Rabusseau[4,5], Alejandro Perdomo-Ortiz[6]**

**1** Zapata AI, Boston, USA
**2** Perimeter Institute for Theoretical Physics, Waterloo, Canada
**3** Institute for Quantum Computing, University of Waterloo, Canada
**4** Mila and DIRO, Université de Montréal, Montréal, Canada
**5** CIFAR AI Chair, Canada
**6** Zapata AI, Toronto, Canada

⋆ jing.chen@zapata.ai , † jacob.miller@zapata.ai

## Abstract

**Beyond their origin in modeling many-body quantum systems, tensor networks have emerged as a promising class of models for solving machine learning problems, notably in unsupervised generative learning. While possessing many desirable features arising from their quantum-inspired nature, tensor network generative models have previously been largely restricted to binary or categorical data, limiting their utility in real-world modeling problems. We overcome this by introducing a new family of tensor network generative models for continuous data, which are capable of learning from distributions containing continuous random variables. We develop our method in the setting of matrix product states, first deriving a universal expressivity theorem proving the ability of this model family to approximate any reasonably smooth probability density function with arbitrary precision. We then benchmark the performance of this model on several synthetic and real-world datasets, finding that the model learns and generalizes well on distributions of continuous and discrete variables. We develop methods for modeling different data domains, and introduce a trainable compression layer which is found to increase model performance given limited memory or computational resources. Overall, our methods give important theoretical and empirical evidence of the efficacy of quantum-inspired methods for the rapidly growing field of generative learning.**

# 1  Introduction

Although originally developed for the needs of quantum many-body physics [1–4], tensor networks (TNs) have rapidly expanded to a host of other areas, where their ability to model correlations and reveal hidden structures within spaces of exponentially large dimension have made them an invaluable tool in such domains as quantum computing [5–7], applied mathematics [8–10], and machine learning (ML) [11–14]. In this last setting, TN models are taken as parameterized models

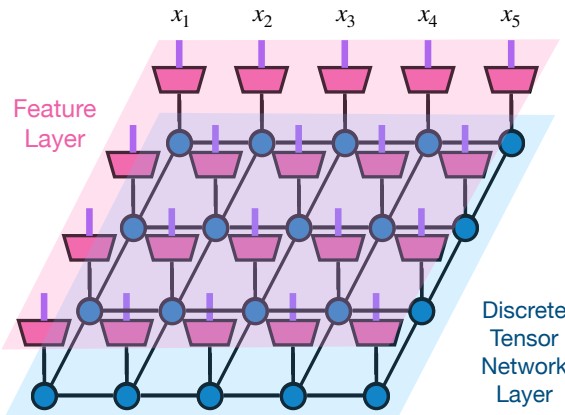

Figure 1: Continuous-valued tensor network. The feature layer (magenta) is a tensor product of feature map operators $\zeta$ defined on each site, with the thicker purple edges denoting indices associated to continuous values. The feature layer is connected to the site indices of a discrete-valued tensor network (blue). The specific network above defines a function $\Phi(\mathbf{x})$, where $\mathbf{x} = (x_1, x_2, \ldots, x_{20})$.

for approximating functions which solve real-world tasks, where TN optimization methods such as the density matrix renormalization group (DMRG) [15] can be repurposed to formulate quantum-inspired approaches to learning the structure of naturally occurring data. This approach has opened up a string of theoretical and empirical successes, from theoretical results in previously intractable problems in learning theory [16,17], to practical high-performance compression methods for large ML models [18,19], to empirical successes across such tasks as image classification [11,20], missing data imputation [21,22], and unsupervised probabilistic modeling [12,23–26].

Generative modeling, where a parameterized model is trained to draw from an unknown probability distribution based on a dataset of previous samples, represents a particularly promising area for the use of TN models in ML [12,23–26]. Beyond the significant intrinsic value of generative modeling for everyday applications (as evidenced by the recent explosion of popular interest in generative AI), TN models using the Born machine (BM) formalism [12,27] present several distinctive benefits within this domain that remain elusive with other classical methods. Some of these benefits arise from the use of tools from entanglement theory, with examples including powerful architecture design methods [28,29] and rigorous expressivity relationships [30,31], while other benefits, such as perfect sampling [32] (and variable-length generalizations [24]), follow from the distinctive mathematical composition of TN models.

TN generative models are not without their limitations however, the most significant of which is their near-exclusive application to distributions of *discrete* random variables. This restriction can be best understood within the BM formalism, which is often thought by physicists as describing many-body wavefunctions. In this context, a TN model can be viewed as a "synthetic" many-body wavefunction, with the number of possible values of each random variable setting the dimension of the associated local spin. Because the BM formalism is primarily used in many-body quantum physics, where a TN describes a discrete "orbital" or site space, it seems natural from that standpoint to use them exclusively for discrete variables. Continuous random variables would necessitate infinite-dimensional local spins, which have received less attention in the many-body TN community. This restriction to discrete random variables severely limits the applicability of TN models in real-world generative modeling, where the majority of problems involve data with continuous features.

In this paper, we present a framework for employing TNs in generative modeling problems involving continuous variables. We make use of vector-valued *feature maps* as a means of map-

ping from the infinite-dimensional space associated with each continuous variable to a finite-dimensional feature space associated with each core of a TN. Using the matrix product state (MPS) ansatz for concreteness, we show how restricting these feature maps to be isometries permits an extension of the standard MPS canonical form to the continuous-valued setting, which in turn allows the use of DMRG update and sweep schemes and perfect sampling algorithms within this new setting. We find that the choice of feature maps and the dimension of the associated feature spaces have a large impact on the behavior of the associated generative model, and develop analytical methods to identify suitable choices of these parameters for different input datasets. Despite the simplicity of this continuous-valued MPS model, which contains the same variational parameters as a standard MPS, we nonetheless prove a *universal approximation theorem* demonstrating that it can approximate any sufficiently smooth probability density function to arbitrary precision, given sufficiently large bond dimensions and feature dimensions. On top of this basic model, we develop a novel *compression layer* that permits the feature map itself to be learned from data, which we show gives significant improvements in the performance of the model for a given number of variational parameters. These methods are empirically evaluated on various synthetic and real datasets containing combinations of discrete and continuous variables, where they are found to reliably capture the features of the dataset in each case.

## 2 Background

Before discussing the continuous case, we first give a brief overview of TNs and BM models in the setting of probability distributions over discrete variables. For a more detailed introduction to TNs and BMs, we refer the interested reader to [4, 12].

### 2.1 Tensor Networks

Tensor networks (TNs) are a general mathematical formalism for representing large multidimensional arrays as the contraction of smaller tensor *cores*. The collection of the model's tensor cores comprise the parameters of the model, whose elements can be varied to achieve high performance in optimization or learning tasks. Because much of the historical development of TNs took place in the setting of condensed matter physics, the multidimensional arrays in question are often thought of by physicists as describing many-body wavefunctions, with the indices of these arrays corresponding to individual spins (e.g. bosons, fermions, or qubits). In machine learning settings though, the tensors in question will describe multivariate functions to be learned from data, in which case the indices will correspond to individual variables, such as those of a multivariate probability distribution. Other use cases of TNs for ML can be found in supervised learning [11, 33], tensor regression [34, 35], and combinatorial optimization [36, 37].

Typical TN models, including all those considered here, use $N$ separate tensor cores $\{A^{(i)}\}_{i=1}^{N}$ to encode an $N$th order tensor $\psi \in \mathbb{K}^{d_1 \times d_2 \times \cdots \times d_N}$, where $\mathbb{K}$ refers to either the real or complex numbers. Each core of the TN contains one *site index* of dimension $d_i$, with the other *bond indices* of $A^{(i)}$ being associated to edges of a graph describing the *network* of tensor contractions connecting the cores of the TN. Different graphs define different TN models, and the graphical structure associated to a TN constrains the correlations achievable between different regions of the model via *area laws* [3, 38]. For a given TN structure, the dimensions of the hidden indices are hyperparameters known as the *bond dimensions* of the model, which determine a trade-off between the expressivity of the model (i.e. the range of tensors which can be represented), and the computational cost of its operation.

Our work utilizes the matrix product state (MPS) model, which is defined by a 1D line graph connecting adjacent sites. The bond dimensions of an MPS can in principle vary for each bond

connecting adjacent sites, but here will be assumed to be some constant value $\chi \geq 1$. In this case, the tensor $\psi$ encoded by the $N$ cores of an MPS is defined by the relation

$$\psi_{\mathbf{s}} = A^{(1)}[i_1] A^{(2)}[i_2] \cdots A^{(N)}[i_N] \tag{1}$$

where $\mathbf{s} = (i_1, i_2, \cdots, i_N)$ is the joint value of all site indices of the tensor, and the RHS of Eq. 1 describes the multiplication of a $\chi$-dimensional row vector $A^{(1)}[i_1]$, $N-2$ different $\chi \times \chi$ matrices $A^{(2)}[i_2] \cdots A^{(N-1)}[i_{N-1}]$, and a $\chi$-dimensional column vector $A^{(N)}[i_N]$. This MPS can therefore be completely described by 2 matrices of dimension $\chi \times d_1$ and $\chi \times d_N$, along with $N-2$ third-order tensors of shape $\{\chi \times \chi \times d_i\}_{i=2}^{N-1}$.

There are typically two different optimization and update strategies. One approach involves updating all tensors incrementally using gradient-based algorithms, as is commonly employed to train neural networks in machine learning settings. The other approach targets one site or two adjacent sites, optimizing them fully before moving to the next target. This method involves inter­actively sweeping and targeting tensors from left to right and then right to left, inspired by DMRG sweeps used in calculating ground states. At each step for a given target, we use gradient descent methods to update the bond tensors until convergence, thereby avoiding the frequent recalculation of environment tensor contractions.

Similar to DMRG schemes, we can target one or two adjacent sites for optimization. In the one-site update approach, the bond dimension is fixed and predetermined. For the two-site update, the two tensors are contracted to form a bond tensor, which is then optimized via gradient-based methods until convergence. The bond tensor can then be factorized back into two adjacent tensors, with the dimension of the newly factorized bond dynamically adjusted based on the singular value spectra occurring in the decomposition. We will refer to this approach as the DMRG two-site scheme in the following discussion. However, unlike traditional DMRG methods for ground state problems, this approach will not involve solving an eigenvalue problem.

## 2.2 Discrete-valued Born Machines

While TNs such as MPS allow the description of arbitrary tensors $\psi$, in the context of probabilistic modeling we would like our models to describe probability distributions, whose entries are non-negative and sum to 1. The Born machine (BM) model represents a natural way of doing so, which also permits the use of concepts from quantum information within the setting of classical probabilistic modeling. A BM is parameterized by a TN describing a "synthetic wavefunction" $\psi$ over the values $\mathbf{s} = (i_1, i_2, \cdots, i_N)$ of the $N$ discrete random variables, where the elements $\psi_{\mathbf{s}}$ of $\psi$ can either be real or complex. In either case, the probability distribution defined by the TN parameterization of $\psi$ is taken to be that given by the Born rule of quantum mechanics, namely

$$P(\mathbf{s}) = \frac{1}{Z} |\psi_{\mathbf{s}}|^2 , \tag{2}$$

where the partition function $Z$ is defined by

$$Z = \sum_{\mathbf{s}} |\psi_{\mathbf{s}}|^2 . \tag{3}$$

Eqs. 2 and 3 guarantee that $P(\mathbf{s}) \geq 0$ for all $\mathbf{s}$, and that $\sum_{\mathbf{s}} P(\mathbf{s}) = 1$, thus ensuring a valid probability distribution. Although the naive summation in Eq. 3 is exponential in the number of variables $N$, for BMs defined over MPS this can be carried out in time $\mathcal{O}(Nd\chi^3)$, where $d = \max_i d_i$. Alternately, TN *canonical forms* can be used to constrain the tensor cores of the MPS to always satisfy $Z = 1$, in which case the evaluation of probabilities in Eq. 2 only has cost $\mathcal{O}(N\chi^2)$.

BMs are often used in the context of *density estimation*, where the goal is to learn a probability distribution $P$ which approximates a target distribution $Q$ using a finite data set $\mathcal{D} = \{\mathbf{s}^{(j)}\}_{j=1}^T$

of $T$ samples from $Q$. A conventional approach for optimizing the TN cores of the BM is by minimizing the Kullback-Liebler (KL) divergence $\text{KL}(Q, P) = \sum_{\mathbf{s}} Q(\mathbf{s}) \log(Q(\mathbf{s})/P(\mathbf{s}))$ between $P$ and $Q$, which is equivalent to minimizing the cross-entropy loss

$$L(Q, P) = -\sum_{\mathbf{s}} Q(\mathbf{s}) \log(P(\mathbf{s})) \approx \frac{1}{T} \sum_{\mathbf{s} \in \mathcal{D}} -\log(P(\mathbf{s})). \tag{4}$$

Although the first summation in Eq. 4 ranges over all possible values of $\mathbf{s}$, leading to an exponential cost with increasing number of variables $N$, the second summation only depends on the size of the dataset $\mathcal{D}$, and can therefore be used to efficiently train the model to minimize Eq. 4. In this form, we will refer to the finite sum on the right of Eq. 4 as the negative log likelihood (NLL) loss of the model on the dataset $\mathcal{D}$. We note that the same functional definitions as above will be used later for defining the KL divergence and NLL loss for probability density functions of continuous random variables $\mathbf{x}$. While the NLL loss is always non-negative for discrete-valued probabilistic models, in the continuous-valued case it is possible for this quantity to become negative for a sufficiently peaked density $P$.

While BMs are trained in a similar manner to other classical probabilistic models, they possess several distinct advantages. Besides being efficient models for density estimation, BMs are also generative models whose underlying TN factorization allows efficient sampling from the exact distribution $P$, without the need for Monte Carlo or other approximate sampling methods. The existence of such *perfect sampling* [32] methods is closely linked to the efficient TN computation of the partition function $Z$ in Eq. 3, and extends to any BM whose underlying TN has an acyclic graph structure [27]. Furthermore, the interpretation of samples from $P$ as outcomes of a projective measurement on the underlying wavefunction $\psi$ permits the application of tools from quantum information within the setting of classical probabilistic modeling, something which has been used as a powerful theoretical tool for characterizing the expressivity of different model families [30, 31], as well as answering model design questions based solely on the underlying dataset $\mathcal{D}$ [29].

## 2.3 Related Work

The notion of feature functions has previously been used in tensor network models, primarily in the context of classification tasks [11, 13, 39], although also with some applications in function regression [40] and generative modeling [41]. As we discuss later, our interpretation of the feature functions as isometric maps permits straightforward conditional generation and training of the continuous-valued BM model. Refs. [42, 43] look at the question of MPS approximations of continuous functions, but where increasingly fine discretizations of the function are approximated using discrete-valued MPS. Ref. [44] shows how to combine a similar style of discretization with certain analytically tractable feature functions. Ref. [45] presents a universal approximation result for functional tensor trains that we use as an important building block in the development of the universal approximation theorems of Sec. 5. Refs. [46, 47] present similar continuous generalizations of tensor train (TT) models, but whose optimization is handled by very different algorithms.

The work of [48] studies density modeling of continuous data (phrased in terms of TTs rather than MPS), with the "squared tensor train density estimation" variation of their model having many similarities to ours. The distinct origin and focus of the TT and MPS communities lead to several important differences between the model of [48] and the one introduced here. While the model of [48] is similarly capable of perfect conditional and unconditional sampling, this requires the computation of explicit marginals that are trivial in our case owing to the use of an MPS canonical form. This use of canonical forms also allows us to optimize the model using a DMRG update and sweep approach, in contrast to updating all tensors simultaneously by the gradient-based optimization used in [48]. Our compression layer architecture is novel, as are the universal approximation results Theorems 2 and 3 proving that continuous-valued MPS permit the approximation of any (sufficiently smooth) wave function or probability density function.

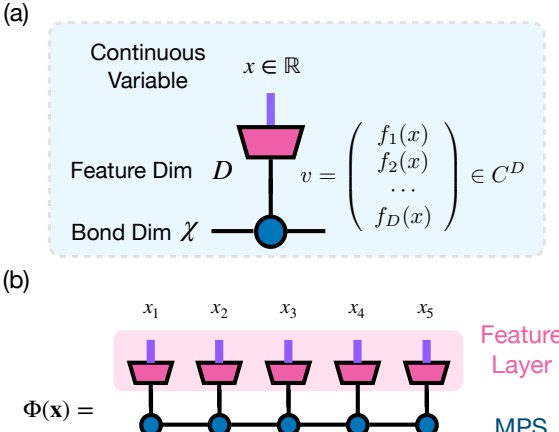

Figure 2: Continuous-valued MPS. (a) For the feature layer, the input at each site $x \in \mathbb{R}$ is a continuous variable, after mapping, it outputs a discrete vector of feature dimension $D$, which is directly connected to the tensor network layer (blue). $\chi$ and $D$ are hyper parameters controlling the dimensions of different bonds. (b) Graphical depiction of the continuous-valued function $\Phi$ defined in Eq. 6.

Our notion of "continuous-valued MPS" should not be confused with the continuous matrix product states introduced in [49]. These models utilize a continuous *spatial* dimension, and can be thought of as the limit of an infinite number of site indices, but with the site dimensions remaining constant and discrete. This type of model has applications in quantum field theory, and is not of interest in this context. The setting we consider here uses a fixed number of indices, but with each index varying over a continuous domain.

## 3    Continuous-valued Born Machine Model

Despite the desirable properties of standard BMs, one obvious limitation is their restriction to modeling discrete-valued probability distributions. While this is rarely an issue in the setting of many-body physics, within classical machine learning such a restriction is extremely limiting, as most datasets used in unsupervised learning contain continuous features described by a probability density function (PDF).

To remedy this limitation, we show here how discrete-valued Born machines can be naturally generalized to the setting of probability distributions over any combination of discrete and continuous variables, as depicted in Fig. 2. This generalization is made possible by the use of *feature maps* which convert points in the continuous domain into finite-dimensional vectors which can be contracted with the underlying discrete-valued TN. We show in detail how this generalization preserves all of the convenient properties and standard algorithms for BMs, including perfect sampling, density evaluation at specific points in the domain, and efficient computations of the partition functions and marginals. For convenience of presentation, in the following we assume the use of identical feature maps for all $N$ sites of the MPS, which are assumed to possess a common feature dimension $D$, but the generalization to site-dependent feature functions is straightforward.

### 3.1    Model

At a high level, the continuous-valued BM introduced here uses a feature map $\zeta : \mathcal{I} \to \mathbb{K}^D$ to convert variables $x$ from a continuous domain $\mathcal{I}$ to real or complex $D$-dimensional vectors $\mathbf{v} = \zeta(x) \in \mathbb{K}^D$. Once such a map has been defined for each site of a discrete-valued MPS, it can

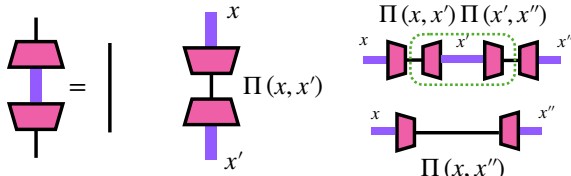

Figure 3: Graphical formulation of the isometric condition on the feature map $\zeta$, which is equivalent to the orthonormality requirement on feature functions expressed in Eq. 6.

be used to convert the MPS into a function of $N$ continuous variables.

A more concrete manner of representing our mapping $\zeta$ comes from picking a basis of $D$ feature functions $\mathcal{F} = \{f_i\}_{i=1}^{D}$, with each $f_i : \mathcal{I} \to \mathbb{K}$ equal to the projection of $\zeta$ onto one of the $D$ vectors $\{e_i\}_{i=1}^{D}$ forming an orthonormal basis for $\mathbb{K}^D$, so that $f_i(x) = \langle e_i, \zeta(x) \rangle$. We require $\zeta$ to be an isometry, meaning that the feature functions satisfy the relations

$$
\begin{aligned}
(\zeta\zeta^\dagger)_{i,j} &= \int_{-\infty}^{\infty} f_i^*(x) f_j(x) dx &&= \delta_{ij}, \\
(\zeta^\dagger\zeta)_{x,x'} &= \sum_{i=1}^{D} f_i(x) f_i^*(x') &&= \Pi(x,x'),
\end{aligned}
\tag{5}
$$

where $\Pi(x,x')$ is a kernel function satisfying $\int_{x'} \Pi(x,x')\Pi(x',x'')dx' = \Pi(x,x'')$ (see Fig. 3). This isometry requirement is invaluable for extending the convenient properties of discrete-valued MPS and BMs to the continuous-valued setting, and can be made without loss of generality, as any feature map can be converted into an isometric form (see Appendix A for details).

Given a mapping $\zeta$ satisfying the above conditions, any tensor $\psi$ containing $N$ discrete indices $\mathbf{s} = (i_1, i_2, \ldots, i_N)$ can be promoted into a continuous-valued function $\Phi$ of $N$ continuous variables $\mathbf{x} = (x_1, x_2, \ldots, x_N)$ by contracting each site index with the corresponding vector $\zeta(x_k)$, as described by

$$
\Phi(\mathbf{x}) = \sum_{i_1, i_2, \ldots, i_N} \left( \prod_{k=1}^{N} f_{i_k}(x_k) \right) \psi_{i_1, i_2, \ldots, i_N}.
\tag{6}
$$

A graphical representation of Eq. 6 is shown in Fig. 2, where the tensor $\psi$ is taken to be given by a discrete-valued MPS.

Just as with discrete-valued BMs in Eq. 2, the continuous-valued BM PDF $P$ is given by the elementwise norm squared of the underlying function $\Phi : \Omega \to \mathbb{K}$,

$$
P(\mathbf{x}) = |\Phi(\mathbf{x})|^2.
\tag{7}
$$

$P(\mathbf{x})$ is clearly non-negative everywhere in its domain of definition $\Omega = \mathcal{I}^N$ and, owing to the isometry conditions of Eq. 6, is guaranteed to satisfy the normalization condition $\int_{\mathbf{x} \in \Omega} P(\mathbf{x})d\mathbf{x} = 1$ whenever the underlying discrete-valued MPS $\psi$ satisfies the condition $\sum_{i_1, \ldots, i_N} |\psi_{i_1, \ldots, i_N}|^2 = 1$. For the case of an unnormalized MPS $\psi$, the normalization factor required to ensure the proper normalization of $\Phi$ is precisely the squared norm of $\psi$, which can be efficiently computed using standard MPS methods. In contrast to the discrete-valued case however, it is possible for the PDF $P$ to take values $P(\mathbf{x}) > 1$ at some points $\mathbf{x} \in \Omega$.

The standard canonical form for discrete-valued MPS can be straightforwardly generalized (with the help of the isometric constraints of Eq. 6) to produce a notion of canonical form for continuous-valued MPS, as shown in Fig. 4. Just as with the usual MPS canonical form, this ensures the proper normalization of the BM distribution $P$ throughout training, and simplifies the computation of gradients and other quantities which typically require $\mathcal{O}(\chi^3)$ time to compute.

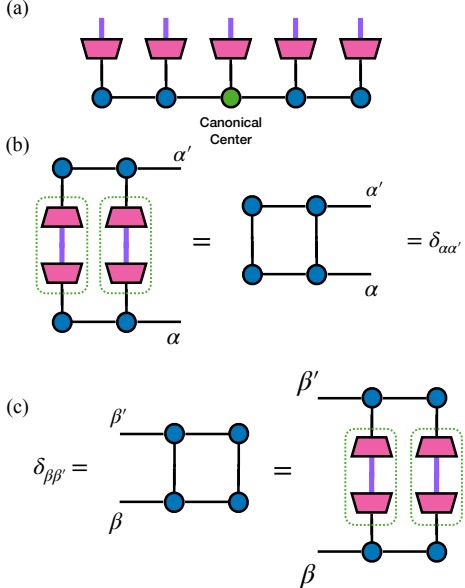

Figure 4: Continuous-valued MPS canonical form. (a) The underlying discrete-valued MPS is required to be in canonical form with an orthogonality center (green dot tensor). When the feature maps additionally satisfy the orthonormality relations of Eq. 6, then the continuous-valued MPS is said to be in continuous-valued MPS canonical form. (b-c) Graphical proof that the left (right) tensors constitute isometries from the left (right) bond spaces to the space of square-integrable functions acting on the left (right) set of continuous variables.

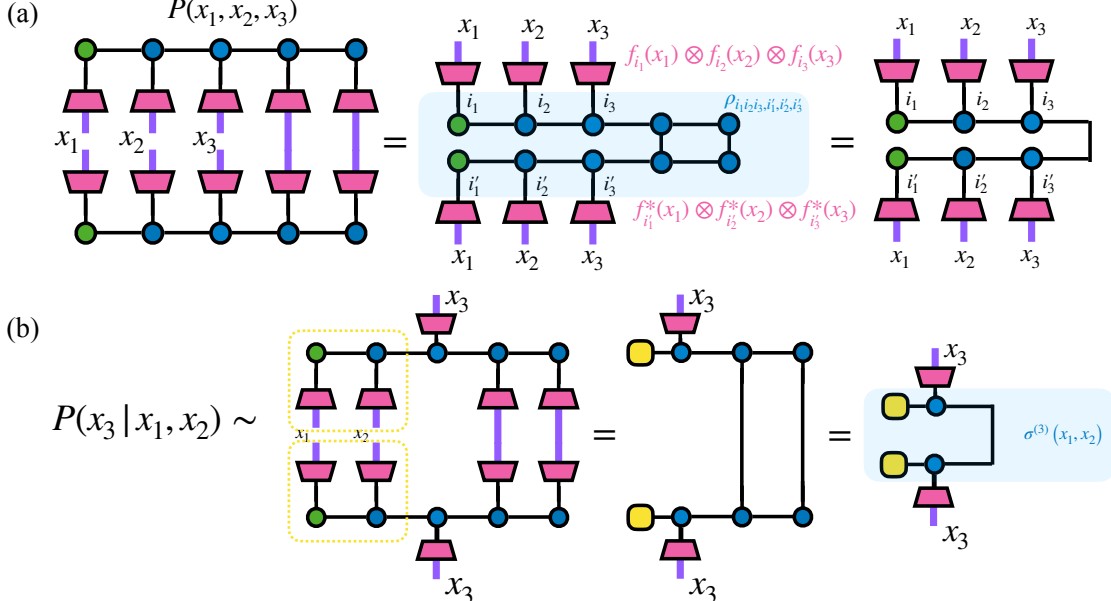

Figure 5: Tensor network diagrams depicting how calculating the probabilities of a continuous-valued MPS BM can be considerably simplified. The MPS is taken to be in canonical form, with the orthonormal center (green dot tensor) on the leftmost site. (a) The marginal distribution $P(x_1, x_2, x_3)$ is given by integrating out the continuous variables $x_4, x_5$, which is trivial when the MPS is in continuous-valued canonical form. (b) The conditional probability $P(x_3|x_1, x_2)$ used in the sampling process, which is facilitated by the computation of a $D \times D$ conditional density matrix $\sigma^{(3)}(x_1, x_2)$.

## 3.2 Sampling

Continuous-valued MPS BMs share the same perfect sampling capabilities as their discrete-valued counterparts. Sampling proceeds site by site, with the continuous random variable at each site $i$ conditioned on those produced at previous sites $1, 2, \ldots, i-1$ via contraction of a sample-dependent vector on the bond dimensions adjacent to site $i$.

For any site $k$, the conditional PDF of the random variable $x_k$ satisfies

$$P(x_k | x_1, x_2, \ldots, x_{k-1}) = \frac{P(x_1, x_2, \ldots, x_{k-1}, x_k)}{P(x_1, x_2, \ldots, x_{k-1})}, \tag{8}$$

where the marginal PDFs are defined for any $k$ as

$$P(x_1, x_2, \ldots, x_k) \tag{9}$$
$$= \sum_{\substack{i_1, \ldots, i_k \\ i'_1, \ldots, i'_k}} \left( \prod_{\ell=1}^{k} f^*_{i_\ell}(x_\ell) \right) \rho_{i_1, \ldots, i_k, i'_1, \ldots, i'_k} \left( \prod_{\ell=1}^{k} f_{i'_\ell}(x_\ell) \right).$$

In the above, $\rho_{i_1, i_2, \ldots, i_k, i'_1, i'_2, \ldots, i'_k}$ represents the discrete reduced density matrix resulting from integrating over all remaining variables $x_{k+1}, \ldots, x_N$. Although the summation involved in Eq. 9, as well as the integrations needed to compute the reduced density matrix, are prohibitively expensive to implement directly, Fig. 5 shows how the tensor network representation of $P$ can be used to remedy this situation.

When the underlying MPS is in canonical form, tracing out the rightmost variables $x_{k+1}, \ldots, x_N$ can be performed efficiently, and computing value of the conditional probability distribution $P(x_k | x_1, x_2, \ldots, x_{k-1})$ can be accomplished with complexity $\mathcal{O}(D^2 \chi^2)$. This process is facilitated by a $D \times D$ conditional density matrix $\sigma^{(k)}(x_1, \ldots, x_{k-1})$ associated to site $k$, shown in Fig. 5b. The conditional distribution in question is then given by

$$P(x_k | x_1, \ldots, x_{k-1}) \tag{10}$$
$$= Z_k^{-1} \sum_{i_k, i'_k = 1}^{D} f^*_{i_k}(x) \sigma^{(k)}_{i_k i'_k}(x_1, \ldots, x_{k-1}) f_{i'_k}(x),$$

where the normalization constant $Z_k$ is chosen such that $\int_{x_k \in \mathcal{I}} P(x_k | x_1, x_2, \ldots x_{k-1}) dx_k = 1$. This in turn can be computed via numerical or closed-form integration over $x_k$, to obtain a cumulative distribution function $F(x_k) = \int_{x' \le x_k} P(x') dx'$. This permits a random sample to be produced using inverse transform sampling, by sampling a uniformly random $z \sim [0, 1]$ and then applying the inverse of the cumulative distribution $F$ to yield the random sample $x_k = F^{-1}(z)$. Continuing this process for $k = 1, 2, \ldots, N$ yields an exact sample from the BM PDF $P(x_1, x_2, \ldots, x_N)$, with $\mathcal{O}\left( N(\chi^2 D + \chi D^2) \right)$ complexity.

## 3.3 Training

In the simplest formulation of a continuous-valued MPS, the feature functions $\mathcal{F} = \{f_i\}_{i=1}^{D}$ are chosen in advance and unchanged throughout training. Only the core tensors of the discrete-valued MPS representation of $\psi$ are taken as tunable parameters, and are trained to minimize the model's NLL on a dataset of unlabeled samples.

Given a dataset with continuous data, each datum can be mapped to a tensor product of vectors associated with the corresponding features at each site. For a dataset $\mathcal{D}$ with $N$ continuous features,

the $j$'th sample $\mathbf{x}^{(j)} = (x_1^{(j)}, x_2^{(j)}, \ldots, x_N^{(j)})$ is mapped into

$$\zeta(x_1^{(j)}) \otimes \zeta(x_2^{(j)}) \otimes \cdots \otimes \zeta(x_N^{(j)}) = \bigotimes_{k=1}^{N} \begin{pmatrix} f_1(x_k^{(j)}) \\ \vdots \\ f_D(x_k^{(j)}) \end{pmatrix}, \tag{11}$$

where $\zeta(x_k^{(j)})$ is the vector representation of the $k$'th feature of the $j$'th sample of $\mathcal{D}$.

Computing the NLL requires a summation over all data samples in $\mathcal{D}$, in each of which the site indices of $\psi$ are contracted with the feature vectors given in Eq. 11. The MPS can then be trained to learn this dataset by any conventional means, such as gradient descent on the NLL of the distribution, or an adapted version of DMRG [12]. In this latter method, the cores for a pair of sites $(i, i+1)$ are trained by first contracting the tensor $\psi$ with the feature vectors at sites $1, 2, \ldots, i-1$ and $i+2, i+3, \ldots, n$, then optimizing the remaining bond tensor to minimize the NLL according to the procedure described in [12].

## 4   Feature Functions

In a setting with discrete data, the possible values of the dataset's categorical features determine the sizes of the site indices of the TN, so that a feature taking $d$ possible values is always associated with a site dimension of $d$. In the continuous-valued setting however, the feature functions and feature dimension $D$ represent new hyperparameters with a significant impact on the inductive bias and expressiveness of the model. The following are all feature maps we assess numerically in Sec. 7, which are natural choices for different types of continuous domains. We describe the component functions of each map, along with their behavior under isometrization (i.e. imposing the isometry conditions of Eq. 6).

**Fourier** The complex exponentials $e^{i2\pi kx}$ for $k = 0, 1, \ldots$ restricted to the compact interval $[0, 1]$, which already satisfy Eq. 6.

**Legendre** Polynomials of degree $k = 0, 1, \ldots$ restricted to the compact interval $[-1, 1]$. Isometrization leads these to be proportional to the Legendre polynomials.

**Laguerre** Polynomials of degree $k = 0, 1, \ldots$ multiplied by the exponential $e^{-x/2}$, and defined on the half interval $\{x \in \mathbb{R} | x \geq 0\}$. Isometrization leads these to be proportional to the Laguerre polynomials multiplied by $e^{-x/2}$.

**Hermite** Polynomials of degree $k = 0, 1, \ldots$ multiplied by the Gaussian $e^{-x^2/2}$, and defined on all of $\mathbb{R}$. Isometrization leads these to be proportional to the Hermite polynomials multiplied by $e^{-x^2/2}$.

Beyond these particular cases, the framework we use permits many other possible feature maps, including the discretization of continuous variables into categorical ones by binning. Consider the $D$ feature functions $f_k$ defined as

$$f_k(x) = \begin{cases} 1, & \lambda_{k-1} \leq x \leq \lambda_k \\ 0, & \text{otherwise.} \end{cases} \tag{12}$$

These indicator functions serve as "one-hot" encodings of the categorical variable associated to the placement of $x$ into one of $D$ separate bins with bin edges $\lambda_0 < \lambda_1 < \cdots < \lambda_D$, and satisfy Eq. 6 up to normalization.

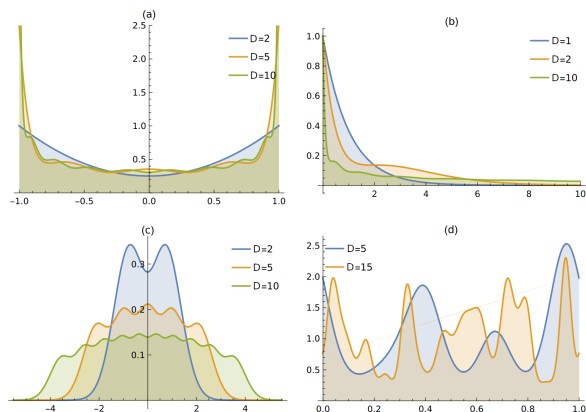

Figure 6: (a-c): The expected univariate distribution $P_{\text{init}}$ when a $D$-dimensional feature map is used. (a) Legendre polynomials, which converge to an arcsin distribution in the limit of $D \to \infty$. (b) Laguerre functions, for which $\mathbb{E}[x]$ increases with increasing $D$. (c) Hermite functions, which progressively broaden with increasing $D$. (d) The expected distribution for Fourier functions is uniform for all $D$, but we instead illustrate the univariate distributions associated with specific random MPS at two values of $D$, for bond dimension $\chi = 5$.

One important and obvious criterion when choosing a feature map is the domain of data being studied. The Lagrange and Fourier feature maps can be used (with appropriate shifting and scaling) to describe data in any connected compact interval $[a, b]$, with the latter also permitting features on a periodic domain (for example, angular data). The Laguerre feature map is suitable for data taking nonnegative real values without an obvious upper limit, while the Hermite feature map is suitable for data which can range over the whole real line.

## 4.1 Priors from Feature Maps

Beyond simply constraining the domain of input data, the choice of feature map for a given continuous-valued BM sets the inductive bias of the model in a manner which can be precisely quantified, in the form of univariate marginal distributions at initialization. A common initialization method for MPS BMs is to choose the elements of each MPS core to be independent identically distributed (IID) random variables, and in this case the following Theorem characterizes the single-site marginal distributions over each continuous random variable.

**Theorem 1.** *Consider a continuous-valued MPS with feature dimension $D$ and an isometric feature map $\zeta : \mathcal{I} \to \mathbb{K}^D$ at site $i$ characterized by feature functions $\mathcal{F} = \{f_1, f_2, \ldots, f_D\}$. Given an initialization of all MPS core elements by IID random variables of zero mean and fixed variance, the expected single-site marginal distribution $P_{init}(x_i)$ of the randomly initialized MPS BM is given by*

$$P_{init}(x_i) = \frac{1}{D}\|\zeta(x_i)\|^2 = \frac{1}{D}\sum_{k=1}^{D}|f_k(x_i)|^2. \tag{13}$$

The proof of Theorem 1 is given in Appendix B, and is based on a simple characterization of the expected density operator of the underlying discrete-valued relative to the IID initialization method in question, which then permits a derivation of Eq. 13.

To illustrate this result, we consider the expected prior distributions associated with each of the features maps considered above. The simplest is the Fourier case, where each complex exponential $f_k(x_i) = e^{i2\pi k x_i}$ will have unit norm, and therefore yield an expected uniform distribution over the interval $\mathcal{I} = [0, 1]$. We note that even in this simple case though, individual random MPS

354 will generally have single-site marginal distributions that differ from this expected distribution
355 (Fig. 6d), which only characterizes the average with respect to many different initializations.

356 More interesting is the Legendre case (Fig. 6a), where the initial distribution skews towards
357 the ends of the interval $\mathcal{I} = [-1, 1]$. In the limit of increasing $D$, the density of the univariate
358 PDF $P_{\text{init}}^{(D)}$ diverges at the endpoints of the interval, yet the distribution as a whole converges to an
359 analytically tractable **arcsin** distribution [50], given by

$$\lim_{D \to \infty} P_{\text{init}}^{(D)}(x) = \frac{1}{\pi \sqrt{1 - x^2}}. \tag{14}$$

360 In practice this bias means the Legendre polynomials lead to significantly worse initialization on
361 most datasets, and we find better performance with other feature maps.

362 The Laguerre and Hermite cases (Fig. 6b and c) are both associated with a broadening of the
363 mass of the expected univariate distribution with increasing $D$, at a rate of $\mathcal{O}(\sqrt{D})$.[1] In this case,
364 it is sensible to rescale the inputs to these feature maps as the feature dimension is increased,
365 i.e. using the new feature functions $g_k(x) = f_k(\sqrt{D}x)$. These rescaled feature functions likely
366 converge to exact analytic forms in the $D \to \infty$ limit, but we leave this characterization as an
367 open question.

368 From a practical standpoint, Theorem 1 represents a useful tool for choosing feature maps
369 based on the marginal distributions associated with each feature of a dataset. Employing a feature
370 map whose expected prior distribution closely resembles the empirical marginal distribution for
371 that feature leads to improved performance in training, in that regions of the feature space which
372 occur more often in the dataset are assigned higher probability at initialization. This could be
373 compared to importance sampling in Monte Carlo methods, which leaves the same asymptotic
374 distribution in the high-capacity limit, but accelerates the rate of convergence.

## 5 Universal Approximation with Continuous-valued MPS

376 It is well-known that discrete-valued MPS with sufficiently large bond dimensions can exactly
377 represent any space of $N$th order tensors using the truncation-free version of the iterated singu-
378 lar value decomposition (SVD) protocol of [5, 9]. By extension, any discrete-valued probability
379 distribution can be exactly represented by an MPS BM whose underlying wavefunction is associ-
380 ated with the square root of the distribution. The corresponding questions for continuous-valued
381 MPS and square-integrable functions (or PDFs) of $N$ continuous variables are considerably less
382 straightforward. It is clear that the exact representation result from the discrete case cannot be
383 applied here, since the continuous-valued functions of interest live in infinite-dimensional Hilbert
384 spaces, while the functions describable by a continuous-valued MPS with fixed bond dimension
385 $\chi$ and feature dimension $D$ will necessarily occupy a finite-dimensional manifold [53].

386 We overcome this difficulty by proving *universal approximation theorems*, which bound the
387 worst-case error in encoding a sufficiently smooth wavefunction (resp. PDF) using a continuous-
388 valued MPS (resp. MPS BM), as a function of the bond dimension $\chi$ and feature dimension $D$.
389 These results show in particular that by increasing the values of $\chi$ and $D$, any sufficiently smooth
390 wavefunction or PDF can be approximated to any desired precision using a continuous-valued
391 MPS.

392 **Theorem 2.** *Consider a family of continuous-valued MPS with polynomial feature functions*
393 $\mathcal{F} = \{f_1, f_2, \ldots\}$ *forming an orthonormal basis for* $[0, 1]$*, which is defined on the hypercube*

---

[1]Hermite distributions have an asymptotic scaling in amplitude as $\left(1 - \frac{x^2}{2D+1}\right)^{-1/2}$ for large $D$ and $|x| \ll \sqrt{2D+1}$, with an exponentially small weight at $|x| \gg \sqrt{2D+1}$ [51]. A similar scaling holds for Laguerre distributions [52].

$\Omega = [0,1]^N \subseteq \mathbb{R}^N$. *Let $k \geq N$ and let $\Phi : \Omega \to \mathbb{C}$ be any square-integrable function with unit norm ($\langle \Phi, \Phi \rangle = 1$), whose partial derivatives of order $1, 2, \ldots, k$ all exist and are bounded. Then for every positive $\chi, D \in \mathbb{N}$ there exists a continuous-valued MPS of bond dimension $\chi$ and feature dimension $D$ with unit norm, whose associated function $\Phi_{\mathrm{MPS}}^{(\chi,D)}$ approximates $\Phi$ with infidelity*

$$1 - |\langle \Phi, \Phi_{\mathrm{MPS}}^{(\chi,D)} \rangle| \leq \gamma_1 \chi^{-k+1} + \gamma_2 D^{-2k}, \tag{15}$$

*where $\gamma_1, \gamma_2 > 0$ depend on the target function $\Phi$, the assumed degree of smoothness $k$, and the feature functions $\mathcal{F}$.*

The proof of Theorem 2, along with an overview of the functional analytic concepts used in the proof and the precise definition of the constants $\gamma_1, \gamma_2$, are given in Appendix C. The result makes heavy use of the work of [45], which generalizes the iterated SVD method for computing discrete MPS representations of tensors to the setting of infinite-dimensional spaces of real-valued functions.

We note that the restriction in Theorem 2, which applies to functions $\Phi$ defined on the unit hypercube $\Omega = [0,1]^N$ is primarily for ease of presentation, and can be easily relaxed to functions on any product of compact intervals $[a_1, b_1] \times \cdots \times [a_N, b_N]$ (i.e. an $N$-dimensional box). More generally, although a rigorous proof for the case of functions on non-compact domains (e.g. $\Phi : \mathbb{R}^N \to \mathbb{C}$) is not possible with the methods of [45], we give a heuristic argument in Appendix C for how Theorem 2 can be modified to bound the error involved in approximating functions on non-compact domains using continuous-valued MPS.

The above theorem can be used to prove a similar approximation result for PDFs. In place of infidelity between wavefunctions, we utilize the Jensen-Shannon (JS) divergence between distributions, which is defined as $\mathrm{JS}(P,Q) = \frac{1}{2}\left(\mathrm{KL}(P,M) + \mathrm{KL}(Q,M)\right)$ for $M$ the equal-weight mixture of $P$ and $Q$ taking values $M(\mathbf{x}) = \frac{1}{2}(P(\mathbf{x}) + Q(\mathbf{x}))$. Besides being symmetric in the input PDFs $P$ and $Q$, JS divergence takes bounded values (in contrast to KL divergence), and is zero only when $P$ and $Q$ are identical almost everywhere. The following Theorem therefore guarantees that any sufficiently smooth PDF can be approximated to arbitrary accuracy using a BM built from a continuous-valued MPS.

**Theorem 3.** *Consider a family of continuous-valued MPS with polynomial feature functions $\mathcal{F} = \{f_1, f_2, \ldots\}$ forming an orthonormal basis for $[0,1]$, which is defined on the hypercube $\Omega = [0,1]^N \subseteq \mathbb{R}^N$. Let $k \geq N$ and let $P : \Omega \to \mathbb{R}$ be any Probability Density Function (PDF) bounded below as $P_{\min} = \min_{\mathbf{x} \in \Omega} P(\mathbf{x}) > 0$, whose partial derivatives of order $1, 2, \ldots, k$ all exist and are bounded. Then for every positive $\chi, D \in \mathbb{N}$ there exists a continuous-valued MPS of bond dimension $\chi$ and feature dimension $D$ with unit norm, whose associated Born machine PDF $P_{\mathrm{MPS}}^{(\chi,D)}(\mathbf{x}) = |\Phi_{\mathrm{MPS}}^{(\chi,D)}|^2$ approximates $P$ with Jensen-Shannon divergence*

$$\mathrm{JS}\left(P_{\mathrm{MPS}}^{(\chi,D)}, P\right) \leq \eta_1 \chi^{-\frac{k-1}{2}} + \eta_2 D^{-k}, \tag{16}$$

*where $\eta_1, \eta_2 > 0$ depend on the target PDF $P$, the assumed degree of smoothness $k$, and the feature functions $\mathcal{F}$.*

The proof of Theorem 3 applies Theorem 2 to the approximation of a naive target wavefunction given by $\Phi_P(\mathbf{x}) = \sqrt{P(\mathbf{x})}$, and then uses standard tools from information theory to translate bounds in infidelity into bounds in JS divergence. The key technical argument of this proof is ensuring that the smoothness guarantees assumed of $P$ yield similar smoothness guarantees for $\Phi_P$, which is complicated by the fact that the derivative of $\sqrt{P(\mathbf{x})}$ becomes infinite in the limit $P(\mathbf{x}) \to 0$. To avoid this pathological behavior, we require that $P(\mathbf{x})$ be bounded below by some $P_{\min}$, as explained in the proof in Appendix C.

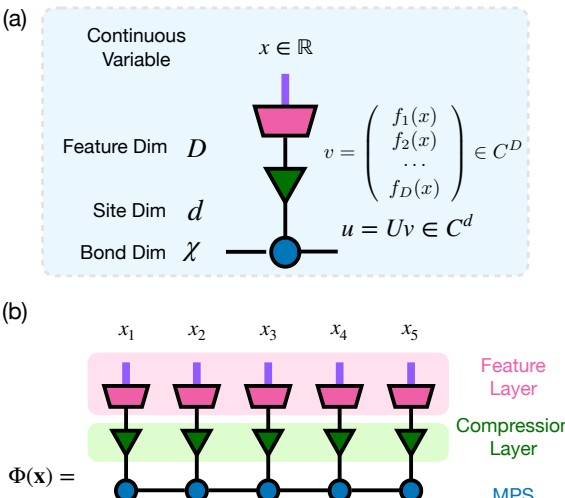

Figure 7: Continuous-valued MPS with compression layer (green). (a) The input $\boldsymbol{x}$ is converted to a vector of dimension $\boldsymbol{D}$. Then the compression operator (green triangular) is an isometry matrix, which rotate and truncate into a $\boldsymbol{d}$ dimensional site index of the MPS. (b) From top to bottom is the feature mapping layer, compression layer and MPS layer. Both the feature layer and the compression layer is direct product of many local operators.

Just as for Theorem 2, the domain in Theorem 3 can be replaced w.l.o.g. with any $\boldsymbol{N}$-dimensional box, and by a heuristic argument can be used to bound the error in approximating PDFs defined on unbounded domains. We note also that different symmetric loss functions can be used in place of JS divergence in Theorem 3, notably total variation distance.

As one final comment on Theorems 2 and 3, the attentive reader might wonder about the case of smooth target functions, for which the value of $\boldsymbol{k}$ can be made arbitrarily large. While the bounds in Eqs. 15 and 16 might seem to become arbitrarily small, it is important to note that the quantities $\boldsymbol{\gamma_1, \gamma_2, \eta_1, \eta_2}$ themselves depend on $\boldsymbol{k}$, and generally grow very rapidly (e.g. super-factorially) with increasing $\boldsymbol{k}$. Consequently, even though smooth PDFs can technically be approximated with error $\mathcal{O}(\boldsymbol{\chi}^{-\frac{k-1}{2}} + \boldsymbol{D}^{-k})$ for any positive value of $\boldsymbol{k}$, in practice the large prefactor in such bounds would make this increasingly favorable scaling only become apparent at values of $\boldsymbol{\chi}$ and $\boldsymbol{D}$ which increase at an astronomical rate.

# 6 Compression Layer

The feature dimension $\boldsymbol{D}$ plays a crucial role in determining the expressivity of the continuous-valued model, as it determines the number of basis functions spanning the space of functions on the continuous variable. This in turn determines the precision of the continuous variable being modeled, with a dimension $\boldsymbol{D}$ limiting the precision to roughly $\mathcal{O}(\boldsymbol{D^{-1}})$. While a larger feature dimension enables the MPS to capture finer details of the distribution, it also comes at the cost of significantly increased computational complexity. As a concrete example, training using two-site update scheme leads to a memory cost of $\mathcal{O}(\boldsymbol{\chi^2 D})$ and a computational cost of $\mathcal{O}(\boldsymbol{\chi^3 D^3})$, making it impractical to increase the feature dimension beyond a certain limit.

To address this issue, we propose the addition of an intermediate compression layer that connects the $\boldsymbol{D}$-dimensional feature space to a smaller site space of dimension $\boldsymbol{d}$ in the underlying discrete-valued MPS. It may be the case in practice that the univariate functions needed to describe each feature of a target distribution or function are easily describable in a low-dimensional

space, but where each basis function is more complex than a predetermined feature function. Our compression layer takes advantage of this possibility by storing a tunable collection of $d$ basis functions, which are each taken to be a superposition of $D$ fixed feature functions, where $d \ll D$. This allows us to take advantage of the expressive power of high numbers of feature functions while minimizing computational costs, improving the efficiency and performance of the continuous MPS model. While we have so far taken $D = d$, when clarity is needed we will refer to $D$ as the feature dimension of the model and $d$ as the site dimension.

Adding a compression layer results in a model that is a simple example of a tree tensor network, as shown in Fig. 7. The compression layer consists of many different $D \times d$ matrices $\{U_i\}_{i=1}^{N}$ satisfying the isometric condition $U_i^{\dagger} U_i = I_d$, which are tunable parameters of the model. In the case of datasets possessing similar kinds of features (e.g. time series data), it may be advantageous to choose all isometries $U_i$ to be equal.

Jointly training the compression layer with the MPS parameters can be done in either the context of gradient-based optimization, or in an alternating manner in the context of DMRG. The former case can be straightforwardly handled by the use of tools for gradient-based optimization on Stiefel manifolds (i.e. families of isometric matrices), so we describe here the latter optimization process. The isometry $U_i$ at a site $i$ is trained to maximize the NLL loss associated to a training dataset $\mathcal{D}$, where samples from the dataset are associated with continuous features $\mathbf{x} = (x_1, x_2, \ldots, x_N)$. For a given sample from $\mathcal{D}$, the $N - 1$ features at all other sites $\mathbf{x}_{\hat{i}} = (x_1, \ldots, x_{i-1}, x_{i+1}, \ldots, x_N)$ are contracted with all cores of the underlying discrete-valued MPS, giving a $d$ dimensional vector $\mathbf{v}_{\neg i}$ at site $i$, while the remaining feature $x_i$ is embedded as a $D$ dimensional vector $\zeta(x_i)$. Given this information, the goal is to find the isometry which minimizes the negative log-likelihood loss over the training dataset, or equivalently:

$$U_i = \operatorname{argmax} \sum_{\mathbf{x} \in \mathcal{D}} \log \left( |\langle \zeta(x_i)| U_i |\mathbf{v}_{\neg i}\rangle| \right). \tag{17}$$

We can find a good $U_i$ by first linearizing the $\log(|\cdot|)$ term, which turns Eq. 17 into a Procrustes problem [54] of linear alignment under an isometric constraint. Procrustes problems can be easily solved with a singular value decomposition on the effective matrix being contracted with $U_i$ in Eq. 17 (after linearization), where setting all singular values to 1 gives the optimal isometry. Upon reaching a candidate solution $U_i$, the nonlinearity $\log(|\cdot|)$ is linearized again and the optimization process repeated until convergence, typically within a few iterations. Full pseudocode for this training is presented in Appendix E.

We note that although computing a vector $\mathbf{v}_{\neg i}$ for each sample $\mathbf{x} \in \mathcal{D}$ may appear expensive, the use of cached environment tensors reduces the incremental cost of this computation to only $\mathcal{O}(\chi^2 d)$ when carried out in the context of the adapted DMRG procedure of [12], making this a very lightweight addition to the basic continuous-valued MPS model.

# 7 Numerical Results

We test the continuous-valued MPS BM model on five distinct density estimation tasks. The first is a rotated hypercube, a simple linearly transformed multidimensional uniform distribution, which we use this to explore the scaling of accuracy with increasing bond and feature dimensions. The second and third are the synthetic two moons dataset [55] and the non-synthetic Iris dataset [56], both of which contain a mixture of continuous and discrete variables. The fourth is a dataset of samples from the classical 2D XY model at nonzero temperature, a statistical mechanical model whose partition function has previously been shown amenable to TN methods [57, 58]. Finally, we use a specifically designed synthetic dataset to test the dynamic basis compression training algorithm.

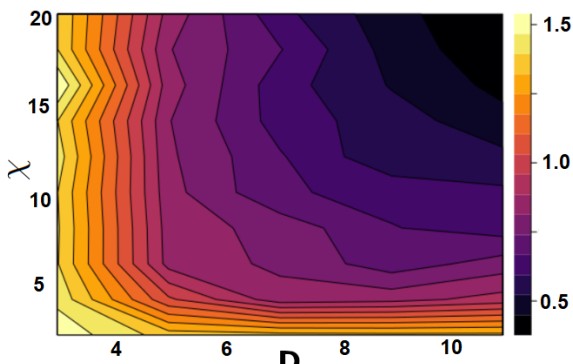

Figure 8: KL divergence of continuous MPS on rotated hypercube dataset, trained with different feature dimensions $D$ and bond dimensions $\chi$.

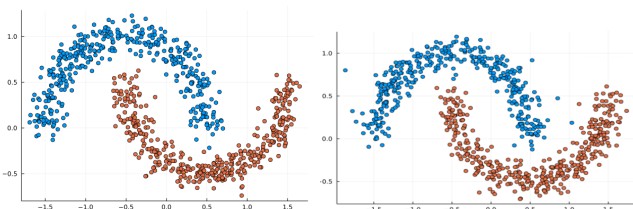

Figure 9: Left plot: 800 samples from the Two Moons distribution, $\sigma = 0.1$. Right plot: 800 samples from our model with $\chi = 8$, $D = 17$.

In each setting the continuous-valued MPS BM is trained to minimize the NLL loss on the target dataset using a two-site DMRG procedure. This is equivalent to minimizing the KL divergence of the model's learned PDF relative to the distribution which produced the target dataset, and in cases where the entropy of the target distribution can be accurately estimated, we will report the KL divergence of the model. Otherwise we report the raw values of the NLL loss, which can be negative in the continuous-valued setting. Any experimental details not specified below can be found in Appendix D.

## 7.1 Rotated Hypercube

As a simple testbed, we used a distribution drawn uniformly from a rotated hypercube $[-1, 1]^N$ for $N = 5$. This dataset has nontrivial correlations between each pair of variables and sharp jumps in the overall density, yet still has continuously differentiable marginals.

For each feature dimension $D$ and maximum bond dimension $\chi$, the MPS was trained from an initial random state with 18 DMRG sweeps and a maximum bond dimension that increased linearly up to $\chi$. The KL divergence of the model for different values of $\chi$ and $D$ make use of the Fourier feature map, which was found to work best in this setting. The KL divergence of the model for different bond and feature dimensions are plotted in Fig. 8. As expected, the loss decreases as we improve either dimension, and saturates if one is increased without the other.

We note that both real and complex tensor networks can be utilized for continuous-valued BMs, and during this initial set of experiments, we quickly found that real-valued tensor networks empirically performed much worse, often failing to converge at all (see Appendix D.1). We attribute this to large jumps in the MPS during the truncation process when using two-site DMRG, and speculate that better behavior might be observed for real-valued models when training using gradient descent. Because of this behavior though, all remaining experiments were carried out using complex-valued tensor networks.

a) True Data

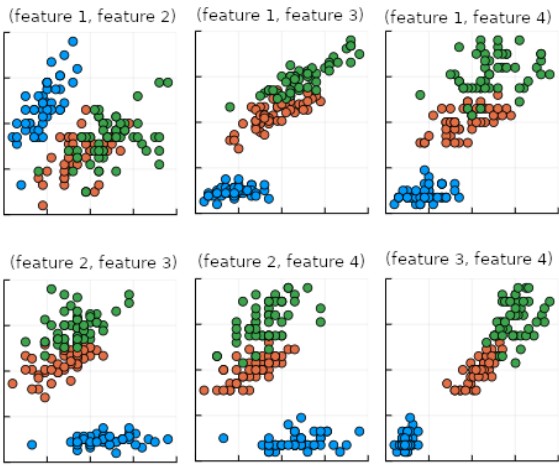

b) Sampled Data

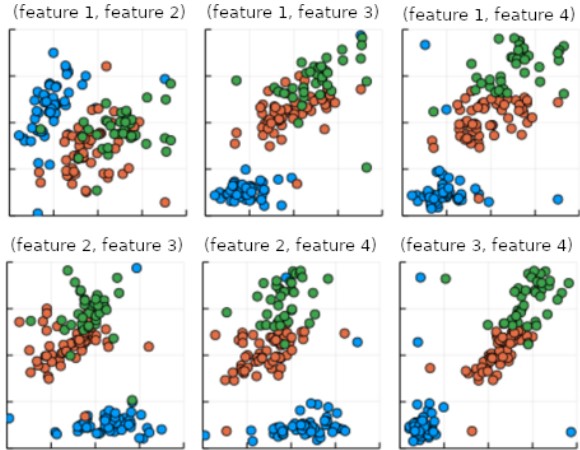

Figure 10: The six different pairwise marginals between each pair of four continuous variables associated to (a) the 150 samples in the Iris dataset, and (b) 150 samples drawn from the continuous MPS model. The three class labels are indicated by color.

## 7.2 Two Moons

The two moons dataset is a standard synthetic dataset available from scikit-learn [55], containing two continuous features encoding the position on a 2D plane and one binary feature indicating which "moon" the sample belongs to. We use a three-site MPS containing two continuous indices and one discrete index to learn the structure of the dataset in an unsupervised manner, but note that the efficient conditional sampling permits the trained MPS BM model to be immediately used as either a supervised classifier or a conditional generative model.

Hermite and Fourier feature maps were both tested on the dataset, with an identical training schedule used as for the rotated cube. We found the Fourier basis to give favorable performance at all parameters, with a comparison of samples from the trained model with those from the two moons distribution shown in Fig. 9. More information, including the KL divergence at different values of $D$ and $\chi$, can be found in Appendix D.2.

### 7.3 Iris Dataset

The Iris dataset [59] has four continuous features and a three-class categorical feature. Being a small dataset of only 150 samples, we must pay attention to overfitting. For each bond dimension and feature dimension under consideration, we use five-fold cross validation, and report the mean of the NLL loss on the validation set in each of the five folds. The petal measurements are strictly positive values, so the Legendre feature map seemed the most natural in this regard. However, we again found that the Fourier feature map performed the best in practice. Although the Iris dataset was used here for an unsupervised density modeling task, we note that as in the two moons task, the MPS BM can immediately be used to either predict the class label given the four continuous features, or conditionally generate continuous samples given a specific class.

We found that overfitting did occur at higher dimensions, with higher losses being seen on the held-out data fold (see Appendix D.3 for the NLL loss as a function of $\chi$ and $D$). Optimal performance was observed at $\chi = 9$ and $D = 7$, with a validation loss of $-1.40 \pm 0.01$. The samples in the Iris dataset are compared to a similar number of samples from the trained MPS BM in Fig. 10, where the four continuous features are displayed as six pairwise marginals. The trained model shows good agreement with the original Iris dataset, although some outliers are visible.

### 7.4 XY Model

The classical XY model [60, 61] is a physical system of 2D unit vectors $\vec{v}_i$, with an interaction energy $E_{i,j} = -\vec{v}_i \cdot \vec{v}_j$ between adjacent sites. For an $N$ site system, representing each vector $\vec{v}_i = (\sin x_i, \cos x_i)$ by its angle $x_i \in [0, 2\pi]$ gives $N$ continous features for each sample drawn from the thermal ensemble associated to the interaction Hamiltonian. This feature space has a natural periodic structure, allowing a further test of the Fourier feature map. We chose $N = 16$, with the associated sites arranged in a $4 \times 4$ grid. To ensure a challenging long-range correlation structure, we trained on a dataset of samples drawn from the model's thermal distribution at temperature $T = 0.8$ which was close to the model's critical temperature of $T_c \approx 0.882$.

The MPS BM model for $\chi = 12$ and $D = 13$ was able to reach a KL divergence of approximately 0.52 relative to the true XY distribution, which was lower than the KL divergence of 0.6 found by a variational autoencoder (VAE) benchmark with hidden dimension of 512 and 10-dimensional latent space. The VAE benchmark additionally required careful hyperparameter tuning and several attempts to reach this value, whereas the continuous-valued MPS was able to reach a lower KL divergence without any modification. Other derived metrics were used to further verify the performance of the MPS model, as reported in Appendix D.4.

### 7.5 Compression Test

To verify that the performance of the compression layer, we created a synthetic dataset containing several tightly-grouped variables (see Appendix D.5 for details). The dataset possesses four continuous features with very different single-site marginals, which are shown in Fig. 11. To assess the impact of the compression layer, we compared three continuous-valued MPS models: (a) a larger MPS model with $D = 16$, (b) a smaller MPS model with $D = 3$, and (c) a compression-enhanced MPS model with distinct feature dimension $D = 16$ and site dimension $d = 3$. Although model (c) employs the same number of feature functions as the larger model (a), its reduced site dimension makes its computational cost closer to the smaller model (b).

The single-site marginal distributions of the three trained models are shown in Fig. 11(a-c), where it is evident that models (a) and (c) give a more faithful reconstruction of the dataset structure than model (b). This is supported by the final NLL loss of the trained models, with model (a) attaining the best NLL loss of -2.17, followed by model (c) with a comparable loss of -2.05, and finally model (b) with a much higher loss of 2.04. We therefore see that by using compression layers, continuous-valued MPS with small site dimensions can deliver performance that is

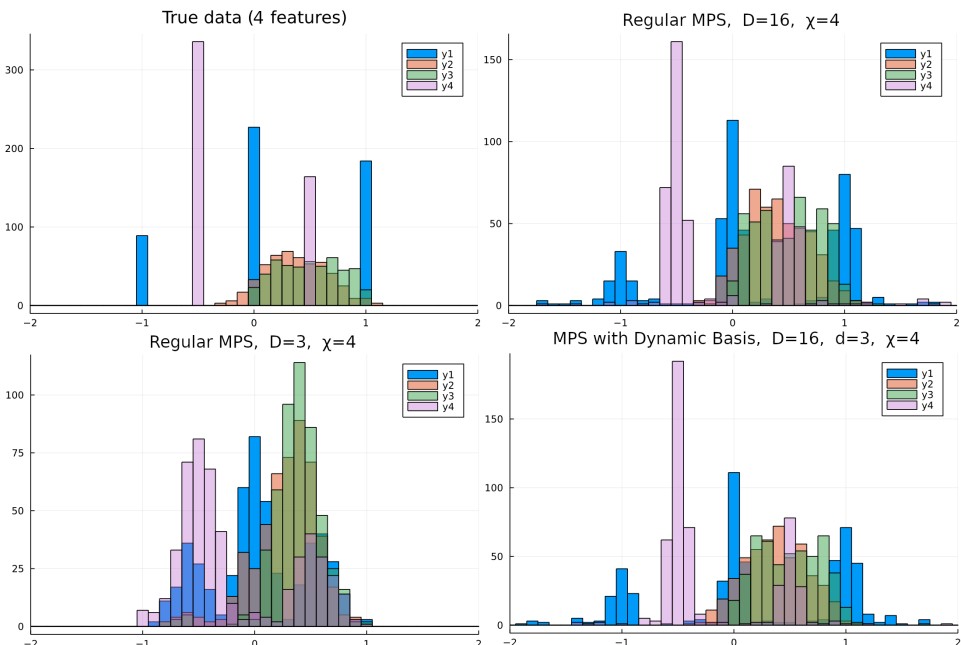

Figure 11: Comparison of the single-feature marginal distributions for three representative MPS models on the synthetic compression dataset, with bond dimension $\chi = 4$. TL: True one-site marginals. TR: an MPS with $D = 16$ obtained the best recovery of the target dataset. BL: The smaller MPS with $D = 3$ gave considerably worse behavior. BT: Using a compression layer with site dimension $d = 3$ and feature $D = 16$ gave comparable performance to the larger model, while maintaining a reduced computational cost.

nearly identical to much larger MPS, but without a significant increase in computational cost or parameter count.

# 8 Conclusions

We have introduced a family of continuous-valued TN generative models, which share the perfect sampling and conditional generation properties of standard discrete-valued TN BMs, while also permitting the use of arbitrary combinations of continuous and discrete data. The generality of these models is proven by a pair of universal approximation theorems, which ensure that any sufficiently smooth PDF or continuous-valued wavefunction can be efficiently represented to arbitrary precision using continuous-valued MPS. Benchmarking this model on a broad range of synthetic and real-world datasets with discrete and continuous variables, we find it able to accurately learn the structure of each dataset, with a programmable compression layer giving enhanced performance in the presence of limited computational resources.

A key ingredient in our continuous generalization is the notion of feature maps to embed continuous data as finite-dimensional vectors. While feature maps have been used for supervised TN models since at least [11, 18], a major contribution of our work is the discovery of much richer structure in feature maps within the context of generative modeling. We prove a general characterization of the influence of feature maps on the marginal distributions of continuous-valued MPS BMs at initialization, and investigate several concrete feature maps in detail from a theoretical and empirical perspective. Our focus on isometric feature maps, which we prove entails no loss of generality, lets us derive a canonical form for continuous-valued MPS that preserves the convenient properties of discrete-valued MPS and permits the use of powerful methods like DMRG for

optimization.

While we have restricted to the use of MPS for convenience, in principle any discrete-valued TN can be extended by our methods into a corresponding continuous-valued model, and benchmarking the performance of more sophisticated TNs (e.g. tree TNs, MERA, and PEPS) in problems with continuous data is an obvious next step. A more open-ended direction is to develop methods for boosting the expressivity of feature maps, or choosing them based on the structure of particular datasets. Our compression layer represents an important contribution along these lines, but using neural networks or other ML models may boost expressivity yet further. Developing heuristics for better choosing the feature dimension $D$ in a given problem, analogous to how entanglement-based area laws guide the choice of bond dimension $\chi$, is another problem deserving future attention. Along similar lines, we anticipate generalizations of two-site update scheme that permit the dynamic variation of both $D$ and $\chi$ to be a useful aid for optimizing continuous-valued TN models.

# 9   Acknowledgments

The authors would like to thank Vladimir Vargas-Calderón for contributing the VAE benchmark result. G.R.'s research was supported by the Canadian Institute for Advanced Research (CIFAR AI chair program).

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

## A  Generality of Isometric Feature Map Condition

Given an arbitrary feature map $\zeta : \mathcal{I} \to \mathbb{K}^D$ represented by a basis of feature functions given by $\mathcal{F} = \{f_1, f_2, \ldots, f_D\}$, we present a general procedure to create new feature functions satisfying Eq. 6, thereby giving an isometric map. We further show how this procedure can be applied to any continuous-valued MPS without imposing any changes in the associated continuous-valued function $\Phi : \mathcal{I}^N \to \mathbb{K}$. We assume first that the functions are linearly independent, as otherwise we can remove any linearly dependent basis functions without any impact on the feature map's expressive power.

First, we calculate a Hermitian "overlap matrix" $M \in \mathbb{K}^{D \times D}$ whose elements $M_{ij}$ give the overlap between feature functions $f_i$ and $f_j$, namely

$$M_{ij} = \langle f_i, f_j \rangle = \int_{x \in \mathcal{I}} f_i^*(x) f_j(x) dx. \tag{18}$$

By the assumption of all functions in $\mathcal{F}$ being linearly independent, $M$ is full-rank, which allows its matrix inverse square root $X = M^{-\frac{1}{2}}$ to be computed from the spectral decomposition of $M$:

$$M = Q \Lambda Q^\dagger \tag{19}$$

$$X = Q \Lambda^{-\frac{1}{2}} Q^\dagger \tag{20}$$

where $Q$ is a $D \times D$ unitary and $\Lambda^{-\frac{1}{2}}$ denotes the elementwise inverse square root of the diagonal matrix $\Lambda$ containing strictly positive diagonal entries. $X$ is itself an invertible Hermitian matrix, and Eq. 19 and Eq. 20 can be used to verify that $XMX = I$

Using $X$ we can generate a new isometric feature map, whose basis of feature functions $\{g_1, g_2, \ldots, g_D\}$ is given by

$$g_k(\mathrm{x}) = \sum_{j=1}^{D} f_j(\mathrm{x}) X_{jk}. \tag{21}$$

The isometric nature of the new feature map can be verified by the orthonormality of the feature functions, concretely:

$$\int_{x \in \mathcal{I}} g_i^*(x) g_j(x) dx$$

$$= \int_{x \in \mathcal{I}} \left( \sum_a f_a^*(\mathrm{x}) X_{ai}^* \right) \left( \sum_b f_b(\mathrm{x}) X_{bj} \right) dx$$

$$= \sum_{a,b} X_{ai}^* X_{bj} \left( \int f_a^*(\mathrm{x}) f_b(\mathrm{x}) dx \right)$$

$$= \sum_{a,b} X_{ai}^* X_{bj} M_{ab} = (X^\dagger M X)_{ij} = \delta_{ij}.$$

Finally, we note that this transformation can be applied to an existing continuous-valued MPS model without any change in the associated function $\Phi$. The new feature map defined by Eq. 21 is equivalent to the composite function sending $x \mapsto \zeta(x)X$, and applying the inverse matrix

Figure 12: (a) Graphical illustration of Lemma 1, characterizing the covariance tensor resulting from IID initialization of an MPS core tensor (blue circles). (b) Simplified graphical proof of Theorem 1 for the case of $N = 5$ and $i = 3$, with the first equality being the definition of the expected marginal distribution $P_{\text{init}}(x_3)$. The second proportionality comes from applying Lemma 1 to all pairs of core tensors and using the isometric nature of the feature maps (magenta trapezoids), with the resulting diagram giving the value $\|\zeta(x_3)\|^2$. The remaining proportionality factor $\frac{1}{D}$ seen in Theorem 1 arises from normalization considerations.

$X^{-1}$ to the corresponding site index of the underlying discrete-valued MPS $\psi$ therefore gives a new discrete-valued MPS which computes the same continuous-valued function $\Phi$, but using an isometric feature map.

Those more familiar with TN methods will identify this procedure as a simple variation of the standard procedure for converting TNs over acyclic graphs into canonical form, but with the important caveat that some of the indices being traced over are associated with infinite-dimensional spaces of functions.

# B Proof of Marginal Distribution Characterization

We prove the characterization of marginal distributions of randomly-initialized continuous-valued MPS BMs given in Theorem 1 in the following, which is restated for ease of reference.

**Theorem 1.** *Consider a continuous-valued MPS with feature dimension $D$ and an isometric feature map $\zeta : \mathcal{I} \to \mathbb{K}^D$ at site $i$ characterized by feature functions $\mathcal{F} = \{f_1, f_2, \ldots, f_D\}$. Given an initialization of all MPS core elements by IID random variables of zero mean and fixed variance, the expected single-site marginal distribution $P_{init}(x_i)$ of the randomly initialized MPS BM is given by*

$$P_{init}(x_i) = \frac{1}{D}\|\zeta(x_i)\|^2 = \frac{1}{D}\sum_{k=1}^{D}|f_k(x_i)|^2. \tag{13}$$

The above Theorem characterizes an initialization coming from choosing each entry of the underlying discrete-valued MPS core tensors $\{A^{(i)}\}_{i=1}^{N}$ to be IID random variables with mean zero and identical variance. In this setting we assume each core tensor $A^{(i)}$ has shape $\chi_{i-1} \times \chi_i \times D_i$, and we begin by proving an important Lemma governing the expected behavior of pairs of such tensors under this type of IID initialization.

811 **Lemma 1.** *Consider a tensor $A^{(i)} \in \mathbb{K}^{\chi_{i-1} \times \chi_i \times D_i}$ whose elements have mean $\mathbb{E}\left[A^{(i)}_{\alpha,\beta,k}\right] = 0$ and*

812 *variance $\mathbb{E}\left[\left|A^{(i)}_{\alpha,\beta,k}\right|^2\right] = t_i$. The sixth-order variance tensor $B^{(i)}$ given by taking two copies of*

813 *$A^{(i)}$ and averaging over all IID initializations is given by*

$$
\begin{aligned}
B^{(i)}_{\alpha,\alpha',\beta,\beta',k,k'} &= \mathbb{E}\left[\left(A^{(i)}\right)^*_{\alpha,\beta,k} A^{(i)}_{\alpha',\beta',k'}\right] \\
&= t \, \delta_{\alpha,\alpha'} \delta_{\beta,\beta'} \delta_{k,k'}.
\end{aligned}
\tag{22}
$$

814   The proof of Lemma 1 is an immediate consequence of the IID nature of the different elements
815 of $A^{(i)}$. The elements of $B^{(i)}$ are covariances between pairs of elements in $A^{(i)}$, which by assump-
816 tion are 0 for different elements, and $t_i$ for identical elements. The result has a convenient graph-
817 ical form, shown in Fig. 12a, which facilitates many calculations involving randomly-initialized
818 MPS.
819   As a concrete example, we can compute the expected squared norm of a continuous-valued
820 MPS $\Phi$ whose underlying discrete-valued MPS $\psi$ has been initialized using core tensors with
821 IID random elements with mean zero. The isometric nature of the model's feature map leads the
822 squared norm of a continuous-valued MPS to be identical to that of its underlying discrete-valued
823 MPS, which with the use of Lemma 1 can be verified to equal the product of all feature and bond
824 dimensions in the model, namely

$$
\mathbb{E}\left[\|\psi\|^2\right] = \prod_{i=1}^{N} t_i D_i \chi_i,
\tag{23}
$$

825 where we take $\chi_N$ to be $\chi_N = 1$. In order to ensure proper normalization when the MPS is used as a
826 probabilistic BM, it is necessary to have the per-core variances $t_i$ to satisfy $\prod_{i=1}^{N} t_i = \prod_{i=1}^{N} D_i \chi_i$,
827 which can be ensured by taking $t_i = (D_i \chi_i)^{-1}$.
828   Given this, the proof of Theorem 1 reduces to taking the definition of the expected univariate
829 marginal distribution $P_{\text{init}}(x_i)$, wherein all other variables are traced out, and applying TN iden-
830 tities to simplify the resultant expression to the form given in Eq. 13 (see Fig. 12b). Applying
831 the isometric condition of Eq. 6 allows all pairs of traced-over feature maps to be removed (see
832 Fig. 3), with Lemma 1 permitting a comparable removal of all pairs of matched tensor cores $A^{(i)}$.
833 The result is the simple diagram on the right side of Fig. 12b, with a proportionality factor equal
834 to the product of all $t_i$ with a term $a_i = \prod_{i=1}^{N-1} \chi_i \prod_{j \neq i} D_j$ coming from tracing over all bond
835 dimensions, as well as all feature dimensions except for $D_i$. The result is the scalar factor $D_i^{-1}$,
836 which under the typical assumption of constant feature dimension $D_i = D$, gives the proportional-
837 ity factor appearing in Eq. 13. This completes the proof of Theorem 1.

# C  Proof of Universal Approximation Results

839 We prove the universality approximation results of Theorems 2 and 3 in the following, which are
840 restated below for ease of reference. In order to prove these Theorems, we must first introduce
841 some concepts from functional analysis, which are used to introduce and prove Theorem 4, which
842 generalizes Theorem 2 to characterize a wider range of functions.

## C.1  Functional Analysis Preliminaries

844 Our results concern the setting of spaces of scalar-valued functions $f : \Omega \to \mathbb{K}$ defined on the $N$-
845 dimensional hypercube $\Omega = [0,1]^N \subseteq \mathbb{R}^N$ equipped with $L_2$-norm $\|f\|_{L^2_\mu} := \left(\int_{\mathbf{x} \in \Omega} |f(\mathbf{x})|^2 d\mu\right)^{1/2}$

associated with a positive-valued finite measure $\mu$ (i.e. $\mu(\Omega) < \infty$). We use $\mathbb{K}$ to indicate one of either $\mathbb{R}$ or $\mathbb{C}$, in the typical case where the choice of field doesn't change the validity of the definitions or results.

If $\mathbf{i} = (i_1, i_2, \ldots, i_k)$ for some $k \geq 0$, then we use $\partial^{(\mathbf{i})} f = \frac{\partial^k f}{\partial x_{i_1} \partial x_{i_2} \cdots \partial x_{i_k}}$ to indicate a standard $k$'th-order partial derivative of the function $f$, with the usual caveat that such derivatives only exist for sufficiently smooth functions. More generally, we use $\mathbf{D}^{(\mathbf{i})} f$ to indicate a $k$'th-order *weak* derivative of $f$, which is defined as a function satisfying the formula

$$\int_{\mathbf{x} \in \Omega} \left( \mathbf{D}^{(\mathbf{i})} f(\mathbf{x}) \right) \varphi(\mathbf{x}) d\mu = (-1)^k \int_{\mathbf{x} \in \Omega} f(\mathbf{x}) \left( \partial^{(\mathbf{i})} \varphi(\mathbf{x}) \right) d\mu \tag{24}$$

for all infinite-differentiable functions $\varphi : \Omega \to \mathbb{K}$ which vanish on the boundary of $\Omega$. The usual integration by parts formula ensures that each partial derivative $\partial^{(\mathbf{i})} f$ is itself a weak derivative of $f$, but the latter can also be defined for functions $f$ whose $k$'th-order partial derivatives don't exist for all $\mathbf{x} \in \Omega$. A function can possess multiple different weak derivatives, but these will agree almost everywhere in $\Omega$ (i.e. everywhere but a measure zero subset of $\Omega$).

We are interested in functions that are "sufficiently nice" for proving universal approximation results, which leads to the concept of *Sobolev spaces*. The $k$'th order Sobolev space $\mathcal{H}_{\mathbb{K}}^k$ on $\Omega$ is defined as the collection of all functions $f : \Omega \to \mathbb{K}$ possessing all weak derivatives $\mathbf{D}^{(\mathbf{i})} f$ of order $|\mathbf{i}| \leq k$ (i.e. $\mathbf{i} = (i_1, i_2, \ldots, i_\ell)$ for $\ell \leq k$), which each have finite $L_2$ norm. This is equivalent to the condition

$$\|f\|_{\mathcal{H}_{\mathbb{K}}^k} := \sum_{|\mathbf{i}| \leq k} \|\mathbf{D}^{(\mathbf{i})} f\|_{L_\mu^2} d\mu < \infty, \tag{25}$$

where the quantity $\|f\|_{\mathcal{H}_{\mathbb{K}}^k}$ is referred to as the $k$'th-order *Sobolev norm* of $f$. For $k = 0$, the Sobolev norm reduces to the usual $L_2$ norm on $\Omega$, and more generally $\|f\|_{L_\mu^2} \leq \|f\|_{\mathcal{H}_{\mathbb{K}}^k}$, so that every $f \in \mathcal{H}_{\mathbb{K}}^k$ necessarily has bounded $L_2$ norm. We will also employ the so-called *Sobolev seminorm* $|f|_{\mathcal{H}_{\mathbb{K}}^k}$ of a function $f \in \mathcal{H}_{\mathbb{K}}^k$, which is defined as $|f|_{\mathcal{H}_{\mathbb{K}}^k} := \sum_{|\mathbf{i}|=k} \|\mathbf{D}^{(\mathbf{i})} f\|_{L_\mu^2}$.

A final concept needed in the following is the notion of $\alpha$-*Hölder continuity*, where a function $f : \Omega \to \mathbb{K}$ is bounded in variation as

$$|f(\mathbf{x}) - f(\mathbf{y})| \leq C \|\mathbf{x} - \mathbf{y}\|^\alpha, \tag{26}$$

for $C \in \mathbb{R}$ a constant holding for any pair of points $\mathbf{x}, \mathbf{y} \in \Omega$. We can without loss of generality take $\alpha$ to be in the range $\alpha \in (0, 1]$, and note that when $\alpha = 1$ the notion of $\alpha$-Hölder continuity reduces to the more familiar definition of Lipschitz continuity. Any function $f : \Omega \to \mathbb{K}$ whose first derivatives exist at all points in $\Omega$ and are bounded as $\left| \frac{\partial f}{\partial x_i} \right| < \infty$ (for $i = 1, 2, \ldots, N$) will always be Lipschitz continuous, and therefore $\alpha$-Hölder continuous for any $0 < \alpha \leq 1$.

## C.2   Proof of Theorem 2

**Theorem 2.** *Consider a family of continuous-valued MPS with polynomial feature functions $\mathcal{F} = \{f_1, f_2, \ldots\}$ forming an orthonormal basis for $[0, 1]$, which is defined on the hypercube $\Omega = [0, 1]^N \subseteq \mathbb{R}^N$. Let $k \geq N$ and let $\Phi : \Omega \to \mathbb{C}$ be any square-integrable function with unit norm ($\langle \Phi, \Phi \rangle = 1$), whose partial derivatives of order $1, 2, \ldots, k$ all exist and are bounded. Then for every positive $\chi, D \in \mathbb{N}$ there exists a continuous-valued MPS of bond dimension $\chi$ and feature dimension $D$ with unit norm, whose associated function $\Phi_{\mathrm{MPS}}^{(\chi, D)}$ approximates $\Phi$ with infidelity*

$$1 - |\langle \Phi, \Phi_{\mathrm{MPS}}^{(\chi, D)} \rangle| \leq \gamma_1 \chi^{-k+1} + \gamma_2 D^{-2k}, \tag{15}$$

881 *where $\gamma_1, \gamma_2 > 0$ depend on the target function $\Phi$, the assumed degree of smoothness $k$, and the*
882 *feature functions $\mathcal{F}$.*

883      Rather than proving Theorem 2 directly, we instead prove a more general Theorem 4, given
884 below. The fact that Theorem 4 implies Theorem 2 is immediate from the definitions and facts
885 above concerning Sobolev spaces and Hölder continuity.

886 **Theorem 4.** *Consider a family of continuous-valued MPS with polynomial feature functions*
887 *$\mathcal{F} = \{f_1, f_2, \ldots\}$ forming an orthonormal basis for $[0, 1]$, which is defined on the hypercube*
888 *$\Omega = [0, 1]^N \subseteq \mathbb{R}^N$. Let $k \geq N$ and let $\Phi : \Omega \to \mathbb{C}$ be any square-integrable function in the*
889 *Sobolev space $\mathcal{H}_{\mathbb{C}}^k$ with unit $L_2$ norm which is $\alpha$-Hölder continuous for $\alpha > \frac{1}{2}$. Then for every*
890 *positive $\chi, D \in \mathbb{N}$ there exists a continuous-valued MPS of bond dimension $\chi$ and feature dimen-*
891 *sion $D$ of unit norm, whose associated function $\Phi_{\mathrm{MPS}}^{(\chi, D)}$ approximates $\Phi$ with infidelity*

$$1 - |\langle \Phi, \Phi_{\mathrm{MPS}}^{(\chi, D)} \rangle| \leq \gamma_1 \chi^{-k+1} + \gamma_2 D^{-2k},$$

892 *where $\gamma_1, \gamma_2 > 0$ depend on the target function $\Phi$, the assumed degree of smoothness $k$, and the*
893 *feature functions $\mathcal{F}$.*

894      This more general formulation allows us to make use of an invaluable result from [45], which
895 applies to functional tensor train (FTT) decompositions that are almost identical to the continuous-
896 valued MPS considered here. The result in question comes from the fundamental FTT approxi-
897 mation characterization given in their Theorem 13 with a polynomial interpolation method, as
898 expressed in their Eqs. 66, 70, and 73 [2]. Rephrased in our terminology and notation, this result
899 takes the form of:

900 **Lemma 2** ( [45]). *Let $\Phi : \Omega \to \mathbb{R}$ be a $\mathcal{H}_{\mathbb{R}}^k$ function on $\Omega = [0, 1]^N \subseteq \mathbb{R}^N$ which is $\alpha$-Hölder*
901 *continuous for $\alpha > \frac{1}{2}$, and where $k \geq N$. Then for any collection $\mathcal{F} = \{f_1, f_2, \ldots\}$ of polynomial*
902 *feature functions which form an orthonormal basis for $[0, 1]$, for every positive $\chi, D \in \mathbb{N}$ there*
903 *exists a continuous-valued MPS with bond dimension $\chi$ and feature dimension $D$ which computes*
904 *a function $\Phi_{\mathrm{MPS}}^{(\chi, D)} : \Omega \to \mathbb{R}$ satisfying*

$$\|\Phi - \Phi_{\mathrm{MPS}}^{(\chi, D)}\|_{L_\mu^2} \leq \sqrt{\frac{N-1}{k-1}} \|\Phi\|_{\mathcal{H}_{\mathbb{R}}^k} (\chi + 1)^{-\frac{k-1}{2}}$$
$$+ C(k) |\Phi_{\mathrm{MPS}}^{(\chi, D)}|_{\mathcal{H}_{\mathbb{R}}^k} D^{-k}, \tag{27}$$

905 *with $C(k)$ depending on $k$ and (implicitly) on the choice of $\mathcal{F}$.*

906      The RHS of Lemma 2 contains two polynomials of $\chi$ and $D$ whose dependence on $k$ is of
907 the same order as the two polynomials on the RHS of the bound of Theorem 4. In order to use
908 the former result to prove the latter though, we must do several things: (a) Replace $\chi + 1$ by $\chi$
909 in the RHS of Eq. 27; (b) Generalize the setting of Lemma 2 from real-valued to complex-valued
910 functions; (c) Ensure that $\Phi_{\mathrm{MPS}}^{(\chi, D)}$ can be chosen to have unit $L_2$ norm whenever $\Phi$ does; (d) Bound
911 the quantity $|\Phi_{\mathrm{MPS}}^{(\chi, D)}|_{\mathcal{H}_{\mathbb{C}}^k}$ by a function of $k$ and $\mathcal{F}$ alone; and (e) Convert the $L_2$ bound derived from
912 Lemma 2 into the infidelity bound appearing in Theorem 4. We will proceed to do each of these
913 in the following.

914 **Replace $\chi + 1$ by $\chi$**     This is straightforward, as for any positive values $K > 0$ and $m \geq 1$,
915 the inequality $K(\chi + 1)^{-m} \leq 2K\chi^{-m}$ holds for all bond dimensions $\chi \geq 1$. This replacment
916 therefore adds a factor of 2 to the first term on the RHS of Eq. 27.

---

[2]All equation and theorem references are relative to the published version of [45]

**Complex-valued generalization**    Although Lemma 2 is phrased in terms of real-valued functions, generalizing this result to complex-valued functions is straightforward. The target function $\Phi^c : \Omega \to \mathbb{C}$ can be represented as a weighted sum of two real-valued functions $\Phi^r, \Phi^i : \Omega \to \mathbb{R}$ via $\Phi^c(x) = \Phi^r(x) + i\Phi^i(x)$, and each function approximated separately by continuous MPS $\Phi^r_{MPS}, \Phi^i_{MPS}$ of bond dimension $\frac{\chi}{2}$ (assuming wlog that $\chi$ is even). The two approximating MPS can be summed together as a single continuous MPS $\Phi^c_{MPS}$ of bond dimension $\chi$, giving

$$
\begin{aligned}
\|\Phi^c - \Phi^c_{MPS}\|_{L^2_\mu} &\leq \|\Phi^r - \Phi^r_{MPS}\|_{L^2_\mu} + \|\Phi^i - \Phi^i_{MPS}\|_{L^2_\mu} \\
&\leq 2\sqrt{\frac{N-1}{k-1}}\left(\|\Phi^r\|_{\mathcal{H}^k_\mathbb{R}} + \|\Phi^i\|_{\mathcal{H}^k_\mathbb{R}}\right)\left(\frac{\chi}{2}\right)^{-\frac{k-1}{2}} \\
&\quad + C(k)\left(|\Phi^r_{MPS}|_{\mathcal{H}^k_\mathbb{R}} + |\Phi^i_{MPS}|_{\mathcal{H}^k_\mathbb{R}}\right)D^{-k} \quad\quad\quad (28) \\
&\leq 2^{\frac{k}{2}+1}\sqrt{\frac{N-1}{k-1}}\|\Phi^c\|_{\mathcal{H}^k_\mathbb{C}}\,\chi^{-\frac{k-1}{2}} \\
&\quad + \sqrt{2}C(k)|\Phi^c_{MPS}|_{\mathcal{H}^k_\mathbb{C}}\,D^{-k}, \quad\quad\quad\quad\quad\quad\quad (29)
\end{aligned}
$$

where we have used the identity $\|\Phi^r\|_{\mathcal{H}^k_\mathbb{R}} + \|\Phi^i\|_{\mathcal{H}^k_\mathbb{R}} \leq \sqrt{2}\|\Phi^c\|_{\mathcal{H}^k_\mathbb{C}}$ (a basic consequence of complex versus real $L_2$ norms) for the Sobolev norm and seminorm.

**Ensure $\Phi^{(\chi,D)}_{\mathrm{MPS}}$ has unit norm**    Theorem 4 not only assumes a target function $\Phi$ with unit norm, but also ensures a continuous MPS approximation with unit norm. This guarantee is not provided by Lemma 2, whose approximating function $\Phi^{(\chi,D)}_{\mathrm{MPS}}$ is not guaranteed to have the same norm as the target $\Phi$. While we can always rescale $\Phi^{(\chi,D)}_{\mathrm{MPS}}$ to have unit norm, we must understand how this impacts the approximation error, something which can be done through inequalities which hold in any normed vector space. Given a target vector $u$ satisfying $\|u\| = 1$, suppose there exists a vector $v$ which approximates $u$ to within distance $\|u - v\|$. The unit vector $\hat{v} = v/\|v\|$ will then necessarily approximate $u$ to within distance

$$
\begin{aligned}
\|u - \hat{v}\| = \|(u - v) + (v - \hat{v})\| &\leq \|u - v\| + \|v - \hat{v}\| \\
&= \|u - v\| + |1 - \|v\|| = \|u - v\| + |\|u\| - \|v\|| \\
&\leq 2\|u - v\|.
\end{aligned}
$$

**Bound $|\Phi^{(\chi,D)}_{\mathrm{MPS}}|_{\mathcal{H}^k_\mathbb{C}}$ as a function of $k$ and $\mathcal{F}$**    We utilize the fact that the spatial dependence of $\Phi^{(\chi,D)}_{\mathrm{MPS}}$ is entirely mediated by the first $D$ polynomial embedding functions from $\mathcal{F}$, which form an orthonormal basis over the finite-dimensional vector space of polynomials with degree less than $D$. This arrangement means that $\Phi^{(\chi,D)}_{\mathrm{MPS}}$ has all partial derivatives of arbitrary order, which can be used to directly compute its Sobolev seminorm $|\Phi^{(\chi,D)}_{\mathrm{MPS}}|_{\mathcal{H}^k_\mathbb{C}}$ without invoking the notion of weak derivative. Given the simple rules for taking partial derivatives of multivariate polynomials, we can see that any single spatial derivative $\frac{\partial}{\partial x_i}$ will preserve the space of polynomials spanned by the $D$ first embedding functions in $\mathcal{F}$, and consequently be equivalent to a $D \times D$ matrix acting on the $i$'th mode of the discrete MPS $\psi^{(\chi,D)}_{MPS}$ underlying the continuous MPS $\Phi^{(\chi,D)}_{\mathrm{MPS}}$. More generally, every partial derivative $\partial^{(i)}$ will be equivalently to a bounded linear operator $M^{(i)}$ acting on the vector space of $N$'th order tensors $\mathbb{C}^{D\times\cdots\times D} \simeq \mathbb{C}^{D^N}$ where $\psi^{(\chi,D)}_{MPS}$ lives.

With these details in place, the seminorm can be explicitly bounded as

$$|\Phi_{\text{MPS}}^{(\chi,D)}|_{\mathcal{H}_{\mathbb{C}}^k} = \sum_{|\mathbf{i}|=k} \|\partial^{(\mathbf{i})}\Phi_{\text{MPS}}^{(\chi,D)}\|_{L_\mu^2} = \sum_{|\mathbf{i}|=k} \|M^{(\mathbf{i})}\psi_{MPS}^{(\chi,D)}\|$$
$$\leq \sum_{|\mathbf{i}|=k} |M^{(\mathbf{i})}| \cdot \|\psi_{MPS}^{(\chi,D)}\| = \sum_{|\mathbf{i}|=k} |M^{(\mathbf{i})}|, \tag{30}$$

where in the second equality we have invoked the orthonormality of the feature functions and the above-remarked equivalence between the action of $\partial^{(\mathbf{i})}$ on continuous MPS and a finite-dimensional matrix $M^{(\mathbf{i})}$ acting on the underlying discrete MPS $\psi_{MPS}^{(\chi,D)}$. The notation $|M^{(\mathbf{i})}|$ refers to the spectral norm (i.e. the largest singular value) of $M^{(\mathbf{i})}$, and the final equality uses the unit norm assumption $\|\Phi_{\text{MPS}}^{(\chi,D)}\|_{L_\mu^2} = \|\psi_{MPS}^{(\chi,D)}\| = 1$. Although the value of the spectral norms $|M^{(\mathbf{i})}|$ will depend on the choice of basis functions $\mathcal{F}$, it is clear that the RHS of Eq. 30 is finite and depends on nothing else besides $k$, giving us the desired bound on $|\Phi_{\text{MPS}}^{(\chi,D)}|_{\mathcal{H}_{\mathbb{C}}^k}$.

**Convert $L_2$ bound to infidelity bound**    Summarizing our results up to this point, we have proved that for any unit-norm target function $\Phi \in \mathcal{H}_{\mathbb{C}}^k$ and an orthonormal basis of polynomial feature functions $\mathcal{F}$, there exist quantities $\gamma_1', \gamma_2' > 0$ depending only on $\Phi$, $k$, and $\mathcal{F}$ for which there exist unit-norm continuous MPS $\Phi_{\text{MPS}}^{(\chi,D)}$ of arbitrary bond dimension $\chi$ and feature dimension $D$ approximating $\Phi$ with $L_2$ error $\|\Phi - \Phi_{\text{MPS}}^{(\chi,D)}\|_{L_\mu^2} \leq \gamma_1' \chi^{-\frac{k-1}{2}} + \gamma_2' D^{-k}$. However Theorem 4 requires a bound on the infidelity $1 - |\langle\Phi, \Phi_{\text{MPS}}^{(\chi,D)}\rangle|$. This can be achieved by the straightforward inequality $1 - |\langle u, v\rangle| \leq \frac{1}{2}\|u - v\|^2$, which holds for any $u$ and $v$ in a normed vector space. Combining this with our $L_2$ bound gives

$$1 - |\langle\Phi, \Phi_{\text{MPS}}^{(\chi,D)}\rangle| \leq \frac{1}{2}\|\Phi - \Phi_{\text{MPS}}^{(\chi,D)}\|_{L_\mu^2}^2$$
$$\leq \frac{1}{2}\left(\gamma_1' \chi^{-\frac{k-1}{2}} + \gamma_2' D^{-k}\right)^2$$
$$\leq \gamma_1 \chi^{-k+1} + \gamma_2 D^{-2k}, \tag{31}$$

where we use the identity $\gamma_1' \gamma_2' \chi^{-\frac{k-1}{2}} D^{-k} \leq \gamma_1' \gamma_2'(\chi^{-k+1} + D^{-2k})$ to simplify the cross-terms arising from the expansion of the square above to arrive at the constants $\gamma_1 := \frac{1}{2}\gamma_1'^2 + \gamma_1'\gamma_2'$ and $\gamma_2 := \frac{1}{2}\gamma_2'^2 + \gamma_1'\gamma_2'$. This gives us our desired infidelity bound, completing our proof of Theorem 4, and by extension Theorem 2.

As a final note, we consider the case where the domain of the target function $\Phi$ is unbounded (e.g. all of $\mathbb{R}^N$). Although the methods of [45] don't apply in this setting (for reasons related to certain functional analytic lemmas used in the proof of Lemma 2), we can instead consider a sequence of approximations of $\Phi$ by functions $\Phi_\epsilon$ supported on boxes $\Omega_\epsilon$ of increasing size, which each approximate $\Phi$ to within a distance of $\epsilon$. By approximating this sequence of functions of bounded domain using Theorem 2, we can approximate our target $\Phi$ to arbitrary precision, albeit at the cost of introducing another $\epsilon$-dependent term into the error bound of Eq. 15. Although this argument leaves some technical details to be worked out, it is clear that in practice this method offers a concrete means of using continuous-valued MPS as universal function approximators for functions on unbounded domains.

## C.3    Proof of Theorem 3

**Theorem 3.** *Consider a family of continuous-valued MPS with polynomial feature functions $\mathcal{F} = \{f_1, f_2, \ldots\}$ forming an orthonormal basis for $[0, 1]$, which is defined on the hypercube*

$\Omega = [0,1]^N \subseteq \mathbb{R}^N$. *Let $k \geq N$ and let $P : \Omega \to \mathbb{R}$ be any Probability Density Function (PDF) bounded below as $P_{\min} = \min_{\mathbf{x} \in \Omega} P(\mathbf{x}) > 0$, whose partial derivatives of order $1, 2, \ldots, k$ all exist and are bounded. Then for every positive $\chi, D \in \mathbb{N}$ there exists a continuous-valued MPS of bond dimension $\chi$ and feature dimension $D$ with unit norm, whose associated Born machine PDF $P_{\mathrm{MPS}}^{(\chi,D)}(\mathbf{x}) = |\Phi_{\mathrm{MPS}}^{(\chi,D)}|^2$ approximates $P$ with Jensen-Shannon divergence*

$$\mathrm{JS}\left(P_{\mathrm{MPS}}^{(\chi,D)}, P\right) \leq \eta_1 \chi^{-\frac{k-1}{2}} + \eta_2 D^{-k}, \tag{16}$$

*where $\eta_1, \eta_2 > 0$ depend on the target PDF $P$, the assumed degree of smoothness $k$, and the feature functions $\mathcal{F}$.*

The proof of Theorem 3 applies Theorem 2 to the artificial wavefunction $\Phi_P(\mathbf{x}) = \sqrt{P(\mathbf{x})}$, which requires first proving that (a) The partial derivatives of $\Phi_P$ of orders $1, 2, \ldots, k$ (where $k \geq N$) all exist and are bounded. With this established, Theorem 2 gives us an approximating wavefunction $\Phi_{\mathrm{MPS}}^{(\chi,D)}$ with a bounded infidelity relative to $\Phi_P$, and we must (b) Convert the infidelity bound between $\Phi_P$ and $\Phi_{\mathrm{MPS}}^{(\chi,D)}$ into a bound on the Jensen-Shannon (JS) divergence between $P$ and $P_{\mathrm{MPS}}^{(\chi,D)}$. We tackle these issues in turn.

**Prove the partial derivatives of $\Phi_P$ of orders $1, 2, \ldots, k$ (where $k \geq N$) all exist and are bounded.** We employ the multivariate version of Faà di Bruno's formula, which is a generalization of the standard chain rule to higher-order partial derivatives, stated here as

**Lemma 3** (Faà di Bruno). *Consider a multivariate function $g : \Omega \to \mathbb{K}$ for $\Omega \subseteq R^N$ whose partial derivatives $\partial^{(\mathbf{i})}g$ up to order $k$ exist and are bounded, as well as a univariate function $h : \mathbb{K} \to \mathbb{K}$ which is $k$-times differentiable within the range of $g$ (i.e. $g(\Omega) \subseteq \mathbb{K}$). Then the partial derivative of the composite function $h \circ g : \mathbf{x} \mapsto h(g(\mathbf{x}))$ wrt the $\ell$ variables $\mathbf{i} = (x_{i_1}, x_{i_2}, \ldots, x_{i_\ell})$ (with $\ell \leq k$) at a point $\mathbf{x} \in \Omega$ is*

$$\partial^{(\mathbf{i})} h(g(\mathbf{x})) = \sum_{\pi \in \Pi} \frac{D^{|\pi|} h}{dy^{|\pi|}}(g(\mathbf{x})) \cdot \prod_{B \in \pi} \partial^{(B)} g(\mathbf{x}), \tag{32}$$

*where (i) $\pi$ runs through the set $\Pi$ of all partitions of the set $\{i_1, i_2, \ldots, i_\ell\}$; (ii) $B \in \pi$ denotes an iteration over all "blocks" of the partition $\pi$; (iii) $\partial^{(B)}$ denotes the partial derivative with respect to all of the variables $x_i$ with $i \in B$; (iv) and $|\pi|$ indicates the number of blocks in the partition $\pi$.*

The details of Eq. 32 are of little interest to us, as we only use it to bound the partial derivatives $\partial^{(\mathbf{i})} \Phi_P$. To this end, we first use the assumption $P(\mathbf{x}) \geq P_{\min}$ and chain rule for the square root function $h(y) : y \mapsto \sqrt{y}$ to show that

$$\left| \frac{D^n h}{dy^n}(g(\mathbf{x})) \right| = \left| \frac{(2n-3)(2n-5)\cdots(-1)}{2^n} g(\mathbf{x})^{-n+\frac{1}{2}} \right|$$

$$\leq 2^n \left( \frac{1}{P_{\min}} \right)^{n - \frac{1}{2}} =: S_{\max}(n). \tag{33}$$

The fact that $S_{\max}(n)$ is an increasing function of $n$ tells us that $S_{\max} := S_{\max}(k)$ is an upper bound for every derivative of $h$ up to order $k$. Denoting the largest partial derivative of $P$ by $G_P := \max_{|\mathbf{i}| \leq k} \max_{\mathbf{x} \in \Omega} \left| \partial^{(\mathbf{i})} P(\mathbf{x}) \right|$, which is finite by assumption (see Theorem 3), we can use Eq. 32 to give the bound

$$\partial^{(\mathbf{i})} \Phi_P(\mathbf{x}) = \partial^{(\mathbf{i})} h(P(\mathbf{x})) \leq \sum_{\pi \in \Pi} S_{\max} \prod_{B \in \pi} \partial^{(B)} P(\mathbf{x})$$

$$\leq \sum_{\pi \in \Pi} S_{\max} \prod_{B \in \pi} G_P \leq \sum_{\pi \in \Pi} S_{\max} (G_P)^k$$

$$\leq S_{\max} (k G_P)^k, \tag{34}$$

1008 which suffices to prove the existence and boundedness of the partial derivatives of $\Phi_P$.

1009 **Convert the infidelity bound between $\Phi_P$ and $\Phi_{MPS}^{(\chi,D)}$ into a bound on the Jensen-Shannon**
1010 **(JS) divergence between $P$ and $P_{MPS}^{(\chi,D)}$.** We proceed in three steps, using the quantum trace dis-
1011 tance and the classical total variation (TV) distance as intermediate quantities. The trace distance
1012 $T(\Phi,\Phi')$ between pure quantum states $\Phi$ and $\Phi'$ takes the form of $T(\Phi,\Phi') := \sqrt{1-|\langle\Phi,\Phi'\rangle|^2}$,
1013 which can be expressed in terms of the infidelity $\mathcal{I}(\Phi,\Phi') = 1-|\langle\Phi,\Phi'\rangle|$ as $T(\Phi,\Phi') = \sqrt{2\mathcal{I}+\mathcal{I}^2}$.
1014 Thus, the infidelity bound of Eq. 31 gives us a bound on our quantum trace distance of interest.

1015  A well-known interpretation of the quantum trace distance between states $\Phi$, $\Phi'$ is a bound
1016 on the classical TV distance $TV(P,P_{MPS}^{(\chi,D)}) = \sup_{A\subseteq\Omega}\left|P(A)-P_{MPS}^{(\chi,D)}(A)\right|$ between any classical
1017 distributions $P_\Phi$, $P_{\Phi'}$ which arise from von Neumann measurements of the corresponding quan-
1018 tum states [62]. Given that the Born machine distributions are precisely those arising from von-
1019 Neumann measurements of the underlying wavefunctions, we have $TV(P,P_{MPS}^{(\chi,D)}) \le T(\Phi_P,\Phi_{MPS}^{(\chi,D)})$,
1020 and thereby a bound on the TV distance,

$$TV(P,P_{MPS}^{(\chi,D)}) = \sup_{A\subseteq\Omega}\left|P(A)-P_{MPS}^{(\chi,D)}(A)\right| \le T(\Phi_P,\Phi_{MPS}^{(\chi,D)})$$
$$\le \sqrt{\frac{3}{2}}\left(\gamma_1'\chi^{-\frac{k-1}{2}}+\gamma_2'D^{-k}\right). \tag{35}$$

1021  Finally, the Jensen-Shannon divergence is known to be bounded by the TV distance, written
1022 as $JS(P,Q) \le \frac{\ln(2)}{2}TV(P,Q)$ which, combined with the above results, give

$$JS(P,P_{MPS}^{(\chi,D)}) \le \frac{\ln(2)}{2}TV(P,P_{MPS}^{(\chi,D)}) \le \frac{\ln(2)}{2}T(\Phi_P,\Phi_{MPS}^{(\chi,D)})$$
$$= \frac{\ln(2)}{2}\sqrt{2\mathcal{I}+\mathcal{I}^2} \le \frac{\sqrt{3}\ln(2)}{2}\sqrt{\mathcal{I}}$$
$$\le \sqrt{\frac{3}{8}}\ln(2)\left(\gamma_1'\chi^{-\frac{k-1}{2}}+\gamma_2'D^{-k}\right). \tag{36}$$

1023 Taking $\eta_1 := \sqrt{\frac{3}{8}}\ln(2)\gamma_1'$ and $\eta_2 := \sqrt{\frac{3}{8}}\ln(2)\gamma_2'$ completes the proof of Theorem 3.

1024 # D Detailed Methods

1025 ## D.1 Rotated Cube

1026 The cube was rotated by a random orthogonal transformation, and then scaled per-axis to the range
1027 $[-1,1]$ to standardize the range. This resulted in a linear transformation

$$M = \begin{bmatrix} 1.33 & 0.155 & 0.074 & 0.411 & 0.029 \\ -0.072 & 1.181 & 0.029 & 0.375 & -0.342 \\ 0.306 & 0.303 & 0.862 & -0.226 & 0.302 \\ -0.363 & 0.217 & -0.297 & 0.998 & 0.125 \\ 0.024 & 0.229 & 0.358 & 0.514 & 0.875 \end{bmatrix}$$

1028 which acted on the set $[-\frac{1}{2},\frac{1}{2}]^5$. The simple form allowed use to compute the exact entropy as
1029 $\log(\det(M)) = -0.4246$.
1030  The training set was 80k sampled points. No minibatching was used. Eighteen sweeps of
1031 DMRG were performed. At each site, 4 steps of gradient descent were performed, each with a

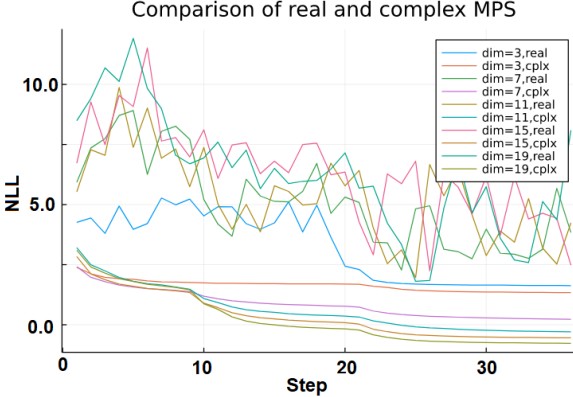

Figure 13: A comparison of real-entry and complex-entry matrix product state performance on the rotated cube dataset. Bond dimension $\chi$ and feature dimension $D$ were equal, and tried at 5 different values from 3 up to 19, with both real and complex entries. The bond dimension was initially lower and increased at training epochs 9 and 22, leading to kinks in the loss curve. Complex-entry MPS trained smoothly, while real-entry MPS did not, due to the sharp truncation of the SVD.

learning rate of **0.05**. The maximum bond dimension in the first sweep was $\max(\chi_{\text{max}}/2, 5)$, and increased in subsequent sweeps linearly up to $\chi_{\text{max}}$ where it stayed for the last five sweeps.

In Fig. 13 we present a comparison of the training performance for MPSs with real and complex entries on this specific data set.

## D.2    Two Moons

For a given noise parameter $\sigma \ll 1$, the entropy of the two moons dataset can be approximated as

$$S \approx \frac{3\ln(2\pi) + 1}{2} + \ln(\sigma) + \frac{1.81}{\pi}\sigma. \tag{37}$$

This approximation can be understood as $\ln(2)$ for choosing a curve to lie on, $\log(\pi)$ for a uniform distribution on a curve of length $\pi$, and $\log(\sigma\sqrt{2\pi e})$ for a radial uncertainty $\sigma$. The final $\frac{1.81}{\pi}\sigma$ accounts for extending the curve of length $\pi$ at the tips by a blur of $\sigma$, where

$$1.81 \approx \int_{-\infty}^{\infty} -\sqrt{2}(1 + \text{erf}(x))\log\left(\frac{1 + \text{erf}(x)}{2}\right) dx. \tag{38}$$

For our experimental results, we used a value of $\sigma = 0.1$, for which $S \approx 0.96$.

To use the Fourier basis, we first scaled the $x$ and $y$ values to the range $[-0.9, 0.9]$. This rescaling adds a small constant factor to the NLL, but this was corrected for when comparing to the true entropy of the distribution. We used a training set of 10k sampled points. The KL divergence as a function of $\chi$ and $D$ is presented in Fig. 14, where a minimum value of 0.022 was reached. It is apparent that for this dataset, the bond dimension quickly saturated its usefulness past $\chi = 4 \sim 5$, with the largest improvement coming from increasing $D$.

## D.3    Iris

We used the Iris dataset available in the UCI Machine Learning Repository [59], consisting of 150 points with four continuous features describing petal shapes of different Iris flowers, supplemented with a categorical feature describing which of three varieties the flower belongs to. The Iris dataset was normalized by rescaling each feature to lie in the range $[-1, 1]$, before applying a feature map to each. The NLL loss for different bond and feature dimensions is shown in Fig. 15.

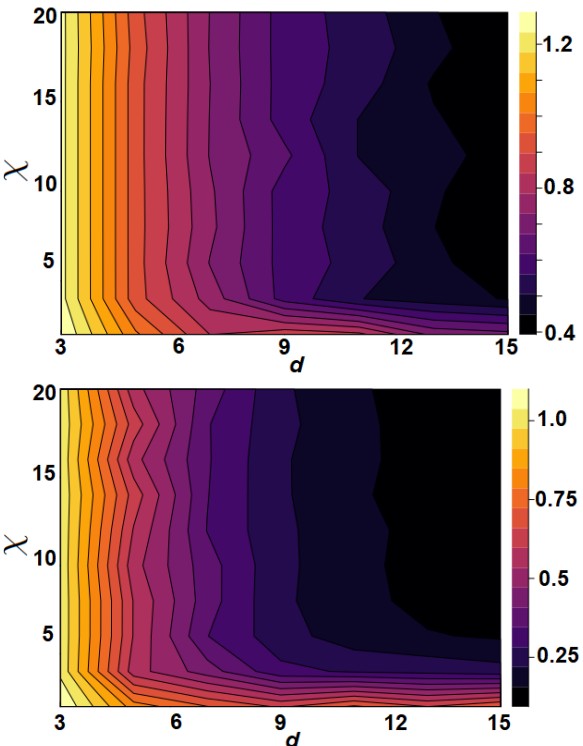

Figure 14: Excess loss (NLL minus distribution entropy) on Two Moons with $\sigma = 0.1$, trained with different embedding dimensions and bond dimensions. Upper plot shows a Hermite embedding. Lower plot shows a Fourier embedding.

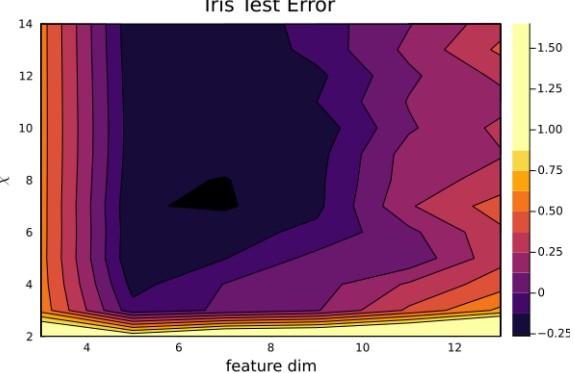

Figure 15: NLL loss on Iris dataset at different bond dimensions and feature dimensions. Each point is the mean of 5 values from a 5-fold cross validation.

### D.4 XY Model

The continuous-valued MPS model was trained to minimize the NLL loss on a dataset of samples drawn from the finite temperature XY model generated using Markov Chain Monte Carlo [63]. The conversion of this score to a KL divergence was made using the test of [64] with 100k samples and a $k = 10$ neighborhood. To further examine the model's behavior, we also measured the KL divergence between the model and true marginal distributions of the angle-invariant quantities $C_{\text{neigh}} = \cos(x_{1,1} - x_{1,2})$ and $C_{\text{corn}} = \cos(x_{1,1} - x_{4,4})$, which measure the correlations between neighbors and opposite corners, respectively. These pairwise correlations were both learned very well, with a KL divergence of 0.0027 for corner-to-corner correlations (which are the hardest for the linear MPS to learn), and even lower values for closer pairs of sites.

### D.5 Compressable Data

The deliberately compressible data for Sec. 7.5 was a 4-feature dataset generated by the following formulas from four samples $x_i$ from the uniform distribution on $[0, 1]$:

$$y_1 = -1 + \lfloor 0.6 + 2.2x_1 \rfloor$$
$$y_3 = x_3$$
$$y_4 = -\frac{1}{2} + \lfloor 1.4x_4 \rfloor$$
$$y_2 = \frac{y_1 + 2x_2 + y_3 + y_4}{4},$$

where $\lfloor x \rfloor$ denotes the largest integer $k$ such that $k \leq x$. This produces a dataset where each feature has a very different marginal (implying that each feature would make best use of a different compression map), the features $y_1$ and $y_4$ are discrete with only three or two values respectively, and $y_2$ is correlated with the other three (so that the MPS correlation structure is not trivial). The single-site marginal distribution is shown in Fig. 11.

## E  Dynamic Basis Training

The following pseudocode details in more detail the process for optimizing the $D \times d$ isometric compression matrices $\{U_i\}_{i=1}^N$ using a dataset of samples $\mathcal{D} = \{\mathbf{x}^{(j)}\}_{j=1}^T$, where each sample has features $\mathbf{x}^{(j)} = (x_1^{(j)}, x_2^{(j)}, \ldots, x_N^{(j)})$. In practice the steps in this process will be interspersed with DMRG updates, in order to benefit from caching of intermediate hidden states.

---

**Algorithm 1** Dynamic Basis Adjustment

---

$\epsilon \leftarrow 0.5$                                                                          ▷ Controls stability
**for** $i \leftarrow 1 \dots N$ **do**                                                             ▷ Loop over each site
    **for** $j \leftarrow 1 \dots T$ **do**                                      ▷ Loop over each sample in batch
        $\mathbf{u}_{i,j} \leftarrow \zeta(x_i^{(j)})$       ▷ $D$-dim embedding
        $\mathbf{v}_{i,j} \leftarrow \text{MPSContract}(\mathbf{x}_{(j)}, \neg i)$      ▷ $d$-dim embedding
        $c_j \leftarrow \mathbf{u}_{i,j}^{\dagger} U_i \mathbf{v}_{i,j}$
        $p_j \leftarrow |c_j|$                                 ▷ Loss probability
        $\phi_j \leftarrow c_j / p_j$                          ▷ Current phase
    **end for**
    $B \leftarrow \sum_{j=1}^{T} (p_j^{\epsilon} \phi_j)^{-1} \mathbf{u}_{i,j} \mathbf{v}_{i,j}^{\dagger}$        ▷ $D \times d$ matrix
    $B_U, B_S, B_V = \text{SVD}(B)$
    $U_i \leftarrow B_U B_V^T$                                                    ▷ Update isometric matrix $U_i$
**end for**
Increase $\epsilon$ towards 1. If loop is unstable, decrease $\epsilon$ towards 0.

---