# Peer review of "Generative Learning of Continuous Data by Tensor Networks"

_SciPost Physics_

## Round 1 · Referee Report · Anonymous (Referee 1) · 2024-5-20

Strengths

1. The context and motivation are clearly presented.
2. The benefits and interesting properties of the proposed method are well articulated.
3. The presented method is well analyzed from a theoretical standpoint.
4. The text is generally well-written and clear.

Weaknesses

1. Lack of comparison between the obtained numerical results and other state-of-the-art methods.
2. Lack of benchmarking on tasks that challenge the performance of state-of-the-art methods.

Report

In this work, the authors propose a modification of tensor network generative models that enables the learning of probability distributions containing continuous random variables. The context, related works, and motivation for their work are presented clearly. The properties of the proposed method—including sampling, training methods, feature maps, and universal approximation—are well-analyzed from a theoretical standpoint. The authors also test the method numerically on five distinct tasks/datasets. In this section, I think it would be beneficial to compare the obtained results with those obtained using other state-of-the-art methods in the literature. Such a comparison with the variational autoencoder is provided in the section about the XY model, but is absent in other sections that present results on more commonly used datasets for benchmarking, such as Two Moons and Iris. Moreover, all benchmarking tasks explored in this work, except for the XY model, could be regarded as toy tasks, which are much simpler than what is accomplished with state-of-the-art distribution learning methods. I understand that the paper is focused on presenting this method and applying it to complex tasks might be out of scope. However, in my opinion, demonstrating the capability of the model on a task closer to real-world scenarios would significantly strengthen the presentation and allow for a better understanding of the practical benefits and limitations of the method.

All in all, I think the work is well-written and presents significant results, so it should be accepted for publication. The proposed improvements are optional.

Requested changes

1. (Optional) A table comparing the performance of the proposed method on benchmark tasks with other methods in the literature.
2. (Optional) An additional section exploring the performance on a more challenging task.

Recommendation

Publish (meets expectations and criteria for this Journal)

  • validity: high
  • significance: high
  • originality: good
  • clarity: high
  • formatting: perfect
  • grammar: perfect

Author:  Jing Chen  on 2024-09-08  [id 4744]

(in reply to Report 1 on 2024-05-20)

Reply to Report 1:

We thank the reviewer for their thoughtful comments and positive evaluation. We agree that a comparison with other state-of-the-art methods and tests on more challenging, real-world tasks would indeed enhance our study. However, given the scope of this initial work, we aimed to establish a foundational framework and its theoretical basis, which naturally limited our focus. We plan to explore these directions in future research, including benchmarking against other methods and applying our model to more complex datasets.

Thank you once again for your valuable feedback, which will certainly guide our ongoing and future work.

---

## Round 1 · Referee Report · Anonymous (Referee 2) · 2024-5-29

Strengths

1. Very clear writing and presentation, with helpful figures and diagrams
2. In-depth examination of certain aspects of the proposed method, like the effect of different choices of basis for the continuous feature maps in the limit of large dimension.
3. Rigorous results about universal approximation properties.
4. Multiple demonstrations on various datasets.
5. Novel idea about adapting the continuous basis during training.

Weaknesses

1. Progress specifically on the algorithm for training the model is somewhat incremental.
2. Demonstrations are on somewhat 'academic' datasets.

Report

The authors present an extension to a framework for unsupervised, generative, modeling using networks. Their extension preserves all of the desirable properties of the underlying tensor network model such as canonical forms and perfect sampling. After describing their proposal and its advantages, they study different possible choices of feature functions and what distributions they approach on average in the limit of many features. They also prove rigorously that the models can capture any function in certain limits, and discuss an interesting compression idea that can be used during training to adapt the feature map. The method is demonstrated to give very good results on five different datasets.

I think this work provides a welcome extension to the framework of using tensor networks for machine learning since continuous features is an important case and needs to be explored carefully. Combined with the clear writing, valuable results, and demonstrations I think this paper should be accepted. I do have a few requested changes below.

Requested changes

The authors assume a quantum physics audience in certain parts of the text, like on page 2 where they say that tensor networks are "often thought of as describing many-body wavefunctions" and on page 1 where they say that the restriction to discrete variables can be best understood through a wavefunction picture. I would strongly suggest that the authors rewrite these sentences a bit to acknowledge this is only one perspective, so they could say "often thought by physicists as describing many-body wavefunctions" on page 2 and on page 1 something like "because the BM formalism is primarily used in quantum physics where a TN describes a discrete 'orbital' or site space, it seems natural from that standpoint to use them exclusively for discrete variables".

I would recommend the authors briefly explain in the text what they mean by "DMRG" since it has slightly different definitions to different readers. Usually it means an algorithm for finding a dominant eigenvector of a linear operator, though the term is certainly overloaded as there is also "time-dependent DMRG" etc. Here I think the authors mainly mean the idea of optimizing two cores at a time and using a low-rank factorization to adapt the bond indices afterward, which is a perfectly good definition.

The authors should correct a small typo on page 3 "REf" instead of "Ref" in the second column.

The authors should consider citing the following papers:
1. "Distributive Pre-Training of Generative Modeling Using Matrix-Product States" arxiv:2306.14787, which proposes similar ideas about continuous feature maps, though used in a different way
2. "Learning multidimensional Fourier series with tensor trains" by Wahls, which includes the idea of continuous inputs to tensor networks though for regression
3. "Generative modeling via tensor train sketching" by Hur et al., which implements a rather different algorithm for generative modeling with tensor networks besides the present one and the cross approximation metioned by the authors
4. "Quantized Fourier and Polynomial Features for more Expressive Tensor Network Models" by Wesel and Batselier, which considers inputting continuous variables by "digitizing" them using the idea of 'quantics' or 'quantized' encoding.

Recommendation

Publish (easily meets expectations and criteria for this Journal; among top 50%)

  • validity: top
  • significance: high
  • originality: high
  • clarity: top
  • formatting: perfect
  • grammar: perfect

Author:  Jing Chen  on 2024-09-08  [id 4745]

(in reply to Report 2 on 2024-05-29)

Reply to Report 2:

Thank you for the valuable feedback.

We have revised the manuscript in the v2 version to address the suggestions, which has significantly improved its readability and clarity.

1. Introduction: We have revised the introduction to make it more accessible to researchers without a background in quantum physics.
2. Clarification of the DMRG Connection: We provided a more precise explanation of how our approach relates to the Density Matrix Renormalization Group (DMRG) scheme and clarified its use in different contexts:
• For sweep optimization strategies, unlike most ML and NN algorithms that update all parameters simultaneously, DMRG optimizes targeted tensors (one-site or two-site) while keeping the environment parameters frozen, optimizing them to their optimal state before moving to the next tensor.
• For the DMRG two-site update scheme, two adjacent tensors are targeted and subsequently factorized into separate tensors with a dynamically adjusted bond dimension and block structure during the sweep.
3. Additional Citations: We have added the four references in Section 2.C (“Related Work”) as suggested, which are highly relevant to our study.
4. Corrections of Typos: We sincerely thank the reviewer for pointing out the typo (“REf” instead of “Ref”) on page 3 and other minor grammatical issues. These have been corrected in the revised version to improve the overall quality of the manuscript.

Regarding the weaknesses identified:

• We acknowledge that the progress on the training algorithm is incremental due to our primary focus on developing the tensor network architecture, particularly the feature mapping and compression layers.
• As an initial exploration, we validated our approach using “academic” datasets to provide a controlled environment for demonstrating the framework’s potential and foundational properties.

Thank you for your thoughtful comments and for helping us improve our manuscript.

---

## Round 1 · Referee Report · Anonymous (Referee 3) · 2024-6-4

Strengths

1. Well and clearly written
2. Useful addition to the literature

Weaknesses

Compression layer could have been discussed in more detail

Report

The authors examine the use of tensor networks (TN) for machine learning with continuous data using a truncated feature map. While this has been proposed before, it has never been made practical. The main contribution of this paper is thus the introduction of the compression layer, which seems a valuable inovation of the feature map formalisme, and several theorems that provide a rigorous basis for the use of their proposed tensor network states. Various numerical illustrations and checks are provided, showing that indeed the approach is practicable.

Requested changes

At the bottom of page 2 the authors state that DMRG would allow them to grow/shrink the bond dimension dynamically, which they say is impossible with gradient descent.
I believe the authors here refer to a 2-site update, the most widely used method for dynamical bond changing. However, his can be done with both gradient descent and DMRG.
Additionally, DMRG (with eigenvalue problem updates for the tensors) is never used for machine learning, only stochastic gradient descent (which differs from DMRG only in the update of the tensor). One could consider this a kind of time evolution, so tDMRG would also be a valid name for it.
I only request this paragraph be changed accordingly

Recommendation

Publish (easily meets expectations and criteria for this Journal; among top 50%)

  • validity: top
  • significance: good
  • originality: high
  • clarity: top
  • formatting: perfect
  • grammar: perfect

Author:  Jing Chen  on 2024-09-08  [id 4746]

(in reply to Report 3 on 2024-06-04)

Reply to Report 3:

We appreciate the positive feedback. In the revised v2 version, we have addressed the suggested clarifications. Details of these changes can be found in the response to Report 2.

We agree that the compression layer is a compelling area for further exploration. However, the primary focus of this study was to establish a comprehensive framework for using tensor networks to handle continuous data, along with a solid mathematical and theoretical foundation. Consequently, the discussion on the compression layer was kept relatively brief as it represents a specific enhancement within the broader framework. We plan to expand on this topic in future work.

Thank you again for your constructive feedback and recommendations, which have helped improve our manuscript.

---

## Round 2 · Referee Report · Anonymous · 2024-9-5

Report

The changes sufficiently address the comments made to the first version.

Recommendation

Publish (easily meets expectations and criteria for this Journal; among top 50%)

---

## Round 2 · Referee Report · Anonymous · 2024-9-10

Report

The changes to this version address all of my comments quite well. I would recommend acceptance at this point.

Recommendation

Publish (easily meets expectations and criteria for this Journal; among top 50%)

---

## Round 2 · Referee Report · Anonymous · 2024-9-17

Report

I believe the updated manuscript is suitable for publication.

Recommendation

Publish (easily meets expectations and criteria for this Journal; among top 50%)

---

## Round 2 · Author Response

We would like to thank the referees for their valuable comments and suggestions. In this revised version of the manuscript, we have made the following changes:

1. Improved Introduction: We have revised the introduction to enhance readability, making it more accessible to researchers without a background in quantum physics.
2. Clarification of the Connection with the DMRG Scheme: We have provided a more precise explanation of the connection between our approach and the Density Matrix Renormalization Group (DMRG) scheme. Additionally, we have distinguished between two scenarios when referring to DMRG to avoid potential misunderstandings:
• In the context of sweep optimization strategies, most machine learning (ML) and neural network (NN) algorithms update all parameters simultaneously. In contrast, the DMRG approach freezes the parameters of the environment and only optimizes the targeted tensors (either one-site or two-site) until they reach their optimal state before moving to the next tensor. The optimization process proceeds from left to right and then from right to left, which is referred to as a “sweep.”
• In the case of the DMRG two-site update scheme, we target two adjacent tensors simultaneously, which are then factorized back into two separate tensors, resulting in a new link index. The bond dimension and block structure of this new link index differ from the original and dynamically adjust during the sweep. In contrast, the one-site update scheme keeps both the bond dimension and block structures fixed during initialization, with updates occurring only to the tensor elements while the structure remains resolved.
3. Additional Citations: We have added four more references in Section 2.C (“Related Work”) as suggested by the referees.
4. Implementation of SciPost LaTeX Template: We have reformatted the manuscript to comply with the SciPost LaTeX template requirements.

---

## Round 2 · List of Changes

1. We have applied the SciPost template in this version.

2. Rephrased the 4th Paragraph of the Introduction:

The original text:
“This restriction can be best understood within the BM formalism, where a TN model can be viewed as a ‘synthetic’ many-body wavefunction, with the number of values obtainable by each random variable setting the dimension of the associated local spin. In this picture, a continuous random variable would necessitate infinite-dimensional local spins, which have received scant attention in the many-body TN community.”

Has been rephrased to:
“This restriction can be best understood within the BM formalism, which is often thought by physicists as describing many-body wavefunctions. In this context, a TN model can be viewed as a ‘synthetic’ many-body wavefunction, with the number of possible values of each random variable setting the dimension of the associated local spin. Because the BM formalism is primarily used in many-body quantum physics, where a TN describes a discrete ‘orbital’ or site space, it seems natural from that standpoint to use them exclusively for discrete variables. Continuous random variables would necessitate infinite-dimensional local spins, which have received less attention in the many-body TN community.”

3. Extended the Last Paragraph of Section III A:

The original text:
“When training an MPS for a given task (e.g. estimating ground state energies, classifying input data, learning probability distributions, etc.), the tensor cores of the MPS can be optimized in several different ways, including gradient descent and density matrix renormalization group (DMRG). We utilize gradient descent in the following for simplicity, but mention that the use of DMRG allows for dynamic control over the bond dimensions of the MPS.”

Has been extended to:
“There are typically two different optimization and update strategies. One approach involves updating all tensors incrementally using gradient-based algorithms, as is commonly employed to train neural networks in machine learning settings. The other approach targets one site or two adjacent sites, optimizing them fully before moving to the next target. This method involves interactively sweeping and targeting tensors from left to right and then right to left, inspired by DMRG sweeps used in calculating ground states. At each step for a given target, we use gradient descent methods to update the bond tensors until convergence, thereby avoiding the frequent recalculation of environment tensor contractions.

Similar to DMRG schemes, we can target one or two adjacent sites for optimization. In the one-site update approach, the bond dimension is fixed and predetermined. For the two-site update, the two tensors are contracted to form a bond tensor, which is then optimized via gradient-based methods until convergence. The bond tensor can then be factorized back into two adjacent tensors, with the dimension of the newly factorized bond dynamically adjusted based on the singular value spectra occurring in the decomposition. We will refer to this approach as the DMRG two-site scheme in the following discussion. However, unlike traditional DMRG methods for ground state problems, this approach will not involve solving an eigenvalue problem.”

4. Adopted More Specific Terminology for the DMRG Approach:

Throughout the paper, we have used more precise terms such as “DMRG update and sweep schemes” when referring to the DMRG approach. Examples include:

- In the last paragraph of Section I:
“… which in turn allows the use of DMRG [update and sweep schemes] and perfect sampling algorithms within this new setting.”

- In the first paragraph of Section IV:
“As a concrete example, training using two-site DMRG [update scheme] leads to a memory cost of …”

- In the last sentence of Section VIII:
“Along similar lines, we anticipate generalizations of two-site DMRG [update scheme] that permit the dynamic variation of both D and χ to be a useful aid for optimizing continuous-valued TN models.”

5. Cited suggested 4 more papers in Section 2.C related work.

---

## Editorial Decision

accepted_in_target_journal